# Galanin neurons in the ventrolateral preoptic area promote sleep and heat loss in mice

Daniel Kroeger[1], Gianna Absi[1], Celia Gagliardi[1], Sathyajit S. Bandaru[1], Joseph C. Madara[2], Loris L. Ferrari[1], Elda Arrigoni[1], Heike Münzberg[3], Thomas E. Scammell[1], Clifford B. Saper [1] & Ramalingam Vetrivelan[1]

The preoptic area (POA) is necessary for sleep, but the fundamental POA circuits have remained elusive. Previous studies showed that galanin (GAL)- and GABA-producing neurons in the ventrolateral preoptic nucleus (VLPO) express cFos after periods of increased sleep and innervate key wake-promoting regions. Although lesions in this region can produce insomnia, high frequency photostimulation of the POA[GAL] neurons was shown to paradoxically cause waking, not sleep. Here we report that photostimulation of VLPO[GAL] neurons in mice promotes sleep with low frequency stimulation (1–4 Hz), but causes conduction block and waking at frequencies above 8 Hz. Further, optogenetic inhibition reduces sleep. Chemogenetic activation of VLPO[GAL] neurons confirms the increase in sleep, and also reduces body temperature. In addition, chemogenetic activation of VLPO[GAL] neurons induces short-latency sleep in an animal model of insomnia. Collectively, these findings establish a causal role of VLPO[GAL] neurons in both sleep induction and heat loss.

[1] Department of Neurology, Program in Neuroscience and Division of Sleep Medicine, Beth Israel Deaconess Medical Center and Harvard Medical School, Boston, MA 02215, USA. [2] Division of Endocrinology, Diabetes and Metabolism, Department of Medicine, Beth Israel Deaconess Medical Center, Harvard Medical School, Boston, MA 02215, USA. [3] Neurobiology of Nutrition and Metabolism, Pennington Biomedical Research Center, Louisiana State University System, Baton Rouge, LA 70808, USA. Correspondence and requests for materials should be addressed to C.B.S. (email: csaper@bidmc.harvard.edu) or to R.V. (email: vramalin@bidmc.harvard.edu)

Almost a century ago, von Economo discovered that patients with injury to the rostral hypothalamus often had unrelenting insomnia[1]. Subsequent animal research showed that lesions of the preoptic area (POA) reduced sleep[2–4] whereas stimulation increased sleep[5–8]. However, the specific circuits constituting this preoptic sleep-inducing area remained unknown. Sherin et al. identified a cluster of neurons in the ventrolateral preoptic nucleus (VLPO) and scattered neurons in the adjacent ventromedial and dorsolateral preoptic area (the extended VLPO) of rats that express cFos during sleep and project to the wake-promoting tuberomammillary nucleus (TMN)[9,10]. About 85% of these neurons contain the inhibitory neurotransmitters galanin (GAL) and GABA[9,10]; similar observations in other species confirmed that most sleep-active neurons in the VLPO and extended VLPO are galaninergic and form a single population, referred to here as the VLPO$^{GAL}$ neurons[11]. Lesions in this region in rodents reduce sleep by 40–50%[12–14]. In humans, loss-of-VLPO$^{GAL}$ neurons is associated with sleep fragmentation and fewer bouts of consolidated sleep[15].

Although GAL is a specific marker for sleep-active neurons in the VLPO and extended VLPO in rats, in mice about 20% of all VLPO$^{GAL}$ cells are active during wake[11]. In addition, a recent optogenetic study found that photostimulation of POA$^{GAL}$ neurons at 10 Hz failed to increase sleep and surprisingly increased wake[16]. These results call into question whether POA$^{GAL}$ neurons promote sleep.

In addition to sleep regulation, the POA is also a key site for thermoregulation[17–19]. Large preoptic lesions cause hyperthermia[3,20], whereas photoactivation of specific POA neurons causes profound hypothermia (4–6 °C)[21–23]. Interestingly, many sleep-active VLPO neurons are also activated by increased skin or body temperature ($T_b$), and may drive reductions in $T_b$, demonstrating the tight coupling of sleep and heat loss pathways[24,25]. However, it is unknown whether VLPO$^{GAL}$ neurons contribute to thermoregulation.

To determine the specific role of VLPO$^{GAL}$ neurons in sleep and thermoregulation, we selectively activated and inhibited VLPO$^{GAL}$ neurons in mice. We find that activation of VLPO$^{GAL}$ neurons promotes sleep even in an animal model of insomnia. Conversely, inhibition of VLPO$^{GAL}$ neurons reduces NREM sleep and increases wake. Interestingly, chemoactivation of VLPO$^{GAL}$ neurons also causes profound hypothermia, indicating that VLPO$^{GAL}$ neurons can also promote heat loss.

## Results

### Distribution of GAL neurons in the POA.
We first investigated the distribution of GAL neurons in the POA of GAL-IRES-Cre:: L10-GFP mice (Methods). In rats and humans, POA$^{GAL}$ neurons are predominantly found in three major clusters, including the VLPO and extended VLPO; the medial preoptic nucleus; and the supraoptic nucleus[11]. We found that in mice, a larger proportion of the VLPO$^{GAL}$ neurons is located in the dorsal extended VLPO. The cluster of GAL neurons in the medial preoptic nucleus is less dense than in rats, but there are many more small (15–20 μm) GAL-expressing neurons that extend along the wall of the third ventricle and into the median preoptic nucleus (Supplementary Fig. 1). There are very few GAL-expressing magnocellular supraoptic neurons at the levels of the POA studied here. Further, the entire POA in a mouse is so small that these cell groups, which are quite distinct in larger species, appear to be crowded together, with a tendency to overlap at the edges. Thus, identifying GAL-expressing cell groups involved in a specific function requires careful histological examination.

### VLPO$^{GAL}$ neurons innervate wake and thermoregulatory centers.
To determine whether VLPO$^{GAL}$ neurons influence brain regions that regulate sleep-wake behavior and $T_b$, we unilaterally microinjected the VLPO of GAL-IRES-cre mice ($n = 8$) with a Cre-dependent adeno-associated viral vector (AAV) coding for channelrhodospsin-2 (ChR2) and mCherry (AAV8-EF1α-DIO-ChR2-mCherry, hereafter 'AAV-ChR2')[26–28]. Six weeks later, we immunolabeled the brain sections for mCherry. We analyzed the projections of 3 cases in which the bulk of the injection involved the three VLPO$^{GAL}$ populations, with little or no involvement of the periventricular, median, or medial preoptic nuclei. Similar to the pattern in rats[10], we found a high density of mCherry-labeled terminals in wake-promoting regions, including the lateral hypothalamic area (LH), TMN, pedunculopontine tegmental nucleus (PPT), lateral and medial parabrachial nucleus (PB) and locus coeruleus (LC). We also observed intense mCherry-labeled terminal fields in the ventrolateral and lateral periaqueductal gray matter (lPAG) and the adjacent lateral pontine tegmentum, a key region for REM sleep regulation[29]. In addition, labeled axons were dense in the dorsal hypothalamic area (DHA) and dorsomedial hypothalamic nucleus (DMH), the raphe pallidus (RPa), and parapyramidal region (PPR) in the rostral medulla, regions essential for thermoregulation[17,18]. These results indicate that VLPO$^{GAL}$ neurons have projections that may allow them to influence many brain regions that regulate both sleep-wake behavior and $T_b$.

### Photoactivation of VLPO$^{GAL}$ neurons increases NREM sleep.
To determine whether VLPO$^{GAL}$ neurons can promote sleep, we injected AAV-ChR2 bilaterally into the VLPO of GAL-IRES-Cre mice ('VLPO$^{GAL}$-ChR2 mice') and implanted bilateral optical fibers in the POA[27] (Fig. 1a). We also placed electrodes for recording the electroencephalogram (EEG) and electromyogram (EMG), and intraperitoneally implanted a telemetry transmitter for recording $T_b$[30,31]. As negative controls, we injected GAL-IRES-cre mice with an AAV without ChR2 (AAV-EF1α-DIO-mCherry, 'AAV-mCherry'). As expected, preoptic injections of either AAV resulted in robust and specific expression of ChR2-mCherry in VLPO$^{GAL}$ neurons, consistent with the distribution of GAL neurons in GAL-IRES-cre::L10-GFP reporter mice (Fig. 1b, c Supplementary Fig. 1). AAV-ChR2 injections in wild-type (WT) littermates did not result in any expression of mCherry.

In vitro, whole cell, current clamp recordings from ChR2-expressing GAL neurons in POA slices indicated that 10 ms blue light pulses at stimulation frequencies ≤2 Hz evoked single action potentials and entrained VLPO neurons in a temporally precise manner with each light pulse evoking a full action potential (crossing 0 mV level) for the entire duration of a 60 s stimulation train (Fig. 2a, b). Similarly, in vivo, low frequency photostimulation (1 Hz, 10 ms duration) for 2 h increased cFos expression in VLPO and extended VLPO ChR2-expressing neurons. Identical stimulation in VLPO$^{GAL}$-mCherry mice did not increase cFos expression (Fig. 1c).

During spontaneous sleep, VLPO neurons typically fire at only 2–5 Hz[25,32], and we found that VLPO$^{GAL}$ neurons cannot entrain to higher frequency photostimulation in vitro. At stimulation frequencies >2 Hz, the peak of the photo-evoked excitatory response, fell below 0 mV, at which point it likely failed to elicit action potentials. While 75% of 4 Hz stimulations triggered action potentials, at 8 and 16 Hz, only the first light pulses evoked an action potential while the following light flashes evoked only excitatory potentials, indicating depolarization block (Fig. 2a, b).

To determine whether photoactivation affects sleep-wake behavior and $T_b$, we photostimulated VLPO$^{GAL}$ neurons

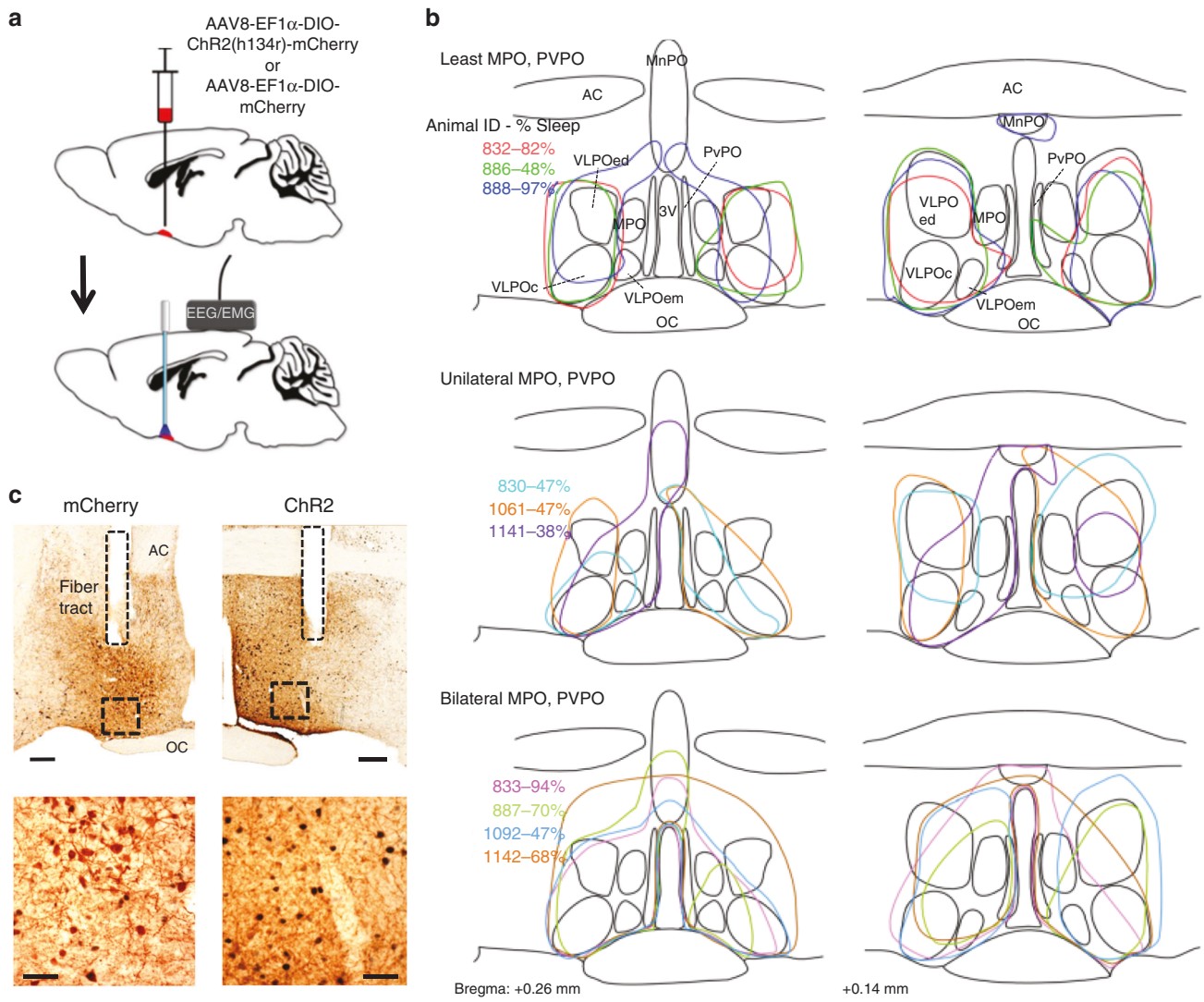

**Fig. 1** Expression of AAV-ChR2 in VLPO^GAL neurons in GAL-IRES-Cre mice. **a** Schematic representation of viral vector injections and implantation of optical fibers and EEG/EMG leads. **b** Outlines of viral injection sites in the VLPO at two levels (equivalent to AP: +0.26 and +AP: 0.14 in Ref. [70]); Animal IDs (n = 10 mice) and the percentage increase in NREM sleep observed after photoactivation in each mouse is represented in matching colors. **c** cFos expression (black nuclei) in mCherry-expressing neurons (brown) after 2 h of optical stimulation (1 Hz, 10 ms) in VLPO^GAL-mCherry mice (left) and VLPO^GAL-ChR2 mice (right). 3 V, 3rd Ventricle; AC anterior commissure; OC optic chiasm; MnPO median preoptic nucleus; MPO medial preoptic area; PvPO periventricular preoptic area; VLPO ventrolateral preoptic area; VLPOc VLPO core; VLPOed VLPO extended dorsal; VLPOem VLPO extended medial. Scale bars in **c** are 200 μm (upper panels) and 50 μm (lower panels)

continuously for 2 h (from 21:00 h to 23:00 h; 2 h after dark onset) and recorded sleep-wake parameters and $T_b$ for 2 h before, during and after stimulation. Consistent with our hypothesis, photo-stimulation at 0.5, 1, 2, and 4 Hz substantially (>60%) increased NREM sleep when compared to sham stimulation in the same mice as well as photostimulation at the same frequencies in VLPO^GAL-mCherry mice (Fig. 2c and Supplementary Table 1), but these stimulations did not affect REM sleep (Supplementary Fig. 2k). Behaviorally, mice with low frequency stimulation (0.5–4 Hz) reduced locomotor activity, assumed a sleep posture within their nest and transitioned into sleep. To determine whether the increase in NREM sleep depended on the duration of light pulses, we photostimulated VLPO^GAL neurons at 1 Hz and varied the duration of light pulses (5, 10, 20, and 50 ms). We found that 10 ms pulses increased NREM sleep slightly more efficiently than either shorter or longer pulses (Supplementary Fig. 2h).

AAV injections in VLPO^GAL-ChR2 mice included GAL neurons in the VLPO (including the dorsal and medial extended VLPO), but also in the medial, median, and periventricular preoptic nuclei in most cases (Fig. 1b). However, a comparison of the different injection sites shows that the mice with the greatest increases in NREM sleep during stimulation (e.g., cases 832, 833, and 888) had injection sites involving the largest part of the VLPO core and the medial and dorsal extended VLPO, whereas mice with lesser increases in sleep (e.g., 830, 1141) typically had less extensive involvement of the VLPO and extended VLPO. In contrast, some mice with large increases in sleep had minimal involvement of the medial, median, or periventricular preoptic GAL neurons (e.g. 832, 888). In addition, in all of these experiments, the optical fiber tips were laterally placed in the dorsal extended VLPO, about 200–400 μm dorsal to the VLPO cluster (Supplementary Fig. 2j). As in our previous work[23], we found that photoactivation resulted in cFos expression only in an area extending about 1 mm ventral to the tip, i.e., the VLPO and extended VLPO neurons, and did not activate neurons in the medial, median, or periventricular preoptic populations.

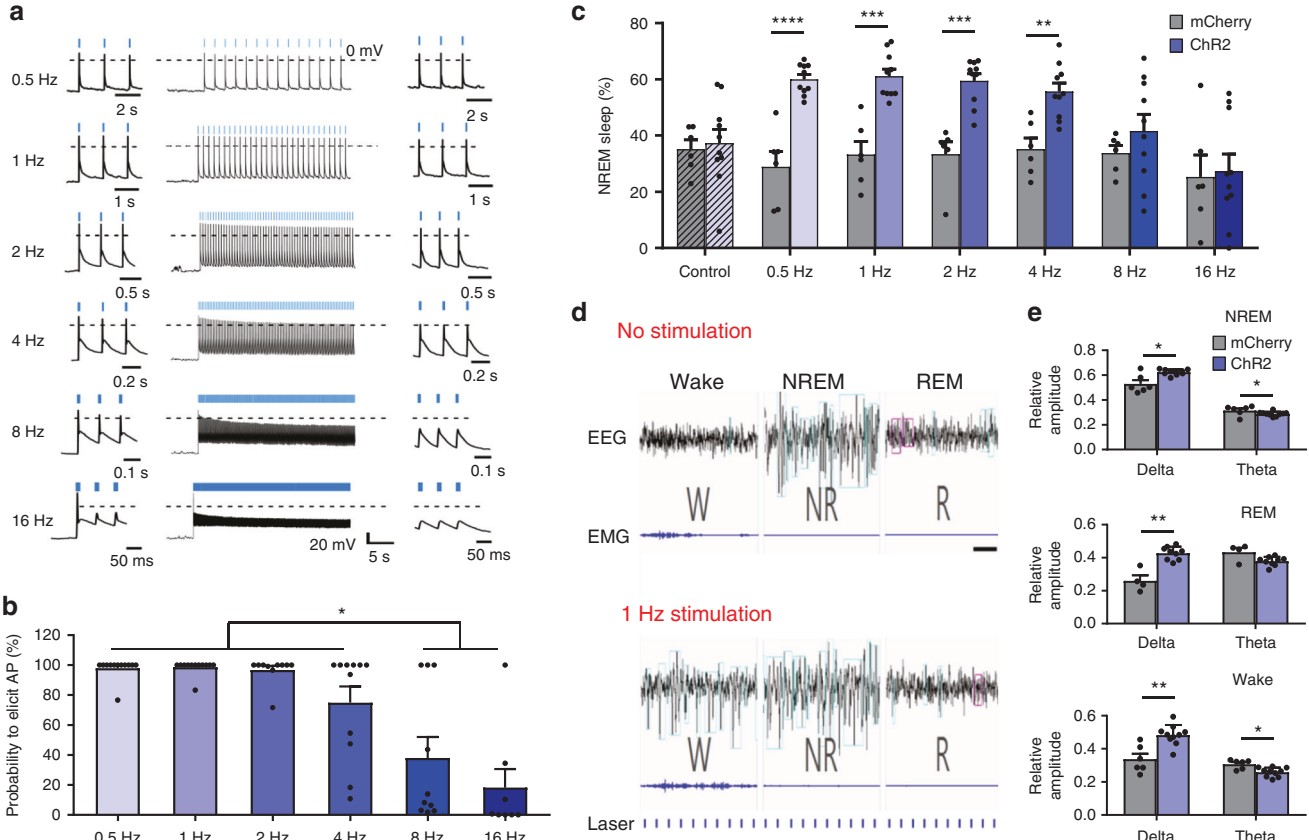

**Fig. 2** Optogenetic activation of VLPO[GAL] neurons increases NREM sleep. **a** In vitro stimulation of ChR2-expressing VLPO[GAL] neurons-one minute recordings with 10 ms pulses at 0.5, 1, 2, 4, 8, 16 Hz (middle column showing the first 30 s of stimulation); first three pulses (left column) and last three pulses (right column); dotted line represents the 0 mV level; laser stimulations in blue above traces. Note that stimulation rates of 8 Hz or above fail to elicit action potentials after the first laser pulses. **b** Quantification of stimulation-induced action potentials over the first minute of stimulation in **a**, one-way ANOVA for treatment, followed by Tukey's post hoc test; $F_{(5, 57)} = 16.11$, $p < 0.0001$ ($n = 13$ cells from three mice). **c** Frequency dependence of NREM sleep with optical stimulation (two-way repeated measures (RM) ANOVA for 2 h of recording time for 'virus type' and 'stimulation frequency' followed by Sidak's post hoc test (virus type: $F_{(1,14)} = 31.69$, $p < 0.0001$; stimulation frequency: $F_{(6,84)} = 5.809$, $p < 0.0001$, $n = 10$ VLPO[GAL]-ChR2 mice vs. $n = 6$ VLPO[GAL]-mCherry mice). **d** Representative sample traces of EEG/EMG recordings during baseline (upper traces) and 1 Hz photostimulation of VLPO[GAL] neurons (lower traces); note large-amplitude and low frequency events in wake, NREM and REM sleep during stimulation; scale bar is 2 s. **e** EEG spectral analysis for NREM sleep, REM sleep and wake during 2 h of 1 Hz stimulation (Mann–Whitney test: NREM (Delta $p = 0.0390$, Theta $p = 0.0390$), REM (Delta $p = 0.0028$, Theta $p = 0.1063$), wake (Delta $p = 0.0017$, Theta $p = 0.0110$), $n = 10$ VLPO[GAL]-ChR2 mice vs. $n = 6$ VLPO[GAL]-mCherry mice). Data are Mean ± SEM. $*p < 0.05$, $**p < 0.01$, $***p < 0.001$, $****p < 0.0001$

We then analyzed sleep architecture and EEG power spectra in mice with 1 Hz stimulations with 10 ms pulse duration, as these parameters increased NREM sleep more efficiently. We found that the photo-evoked increase in NREM sleep was primarily due to more NREM sleep bouts rather than longer bouts (Supplementary Table 1), suggesting a role for VLPO[GAL] neurons in facilitating wake-to-NREM transitions. Interestingly, the EEG showed high voltage slow waves entrained to the frequency of stimulation, or its harmonics (i.e., 1, 2, and 4 Hz; Supplementary Fig. 3) during wake, NREM sleep and REM sleep, resulting in an apparent increase in delta power (Fig. 2d, e and Supplementary Table 1). Identical light pulses in VLPO[GAL]-mCherry mice did not affect the EEG spectrum (Supplementary Table 1). The mechanism for entraining cortical EEG to VLPO firing may involve rhythmic inhibition of wake-promoting neurons.

Although, we observed slow waves entrained to the photo-stimulation frequency during wake and REM sleep (Fig. 2d), the different states were still clearly distinguishable both electro-physiologically and on video. Moreover, the increase in NREM sleep during photostimulation of VLPO[GAL] neurons was followed by a period of 'rebound-wake', i.e. an increase in wake across the

2 h after stimulation (Supplementary Fig. 2a-g), suggesting that the induced sleep fulfilled the homeostatic qualities of natural sleep. Collectively, these observations indicate that VLPO[GAL] neurons promote NREM sleep mainly by initiating sleep and increasing transitions from wake into NREM sleep.

Importantly, and in contrast to low frequency stimulation, higher frequency photostimulation (8 and 16 Hz) of VLPO[GAL] neurons did not increase NREM sleep (Fig. 2c). Our in vitro recordings showed depolarization block with high frequency stimulations, and in vivo, 16 Hz photostimulation actually increased wakefulness and exploratory activity similar to a prior study[16] in 7 out of 10 mice (three mice which did not display wake after 8 or 16 Hz stimulations had comparatively smaller preoptic injections). Thus, high frequency photostimulation of VLPO[GAL] neurons may induce depolarization block, disinhibiting the arousal systems and paradoxically increasing wakefulness.

Finally, although low frequency (1 Hz) stimulation slightly decreased $T_b$ (~0.5 °C), overall, neither low nor high frequency stimulations altered $T_b$ during the 2 h recording period when compared to sham stimulation in the same mice (Supplementary Fig. 2i).

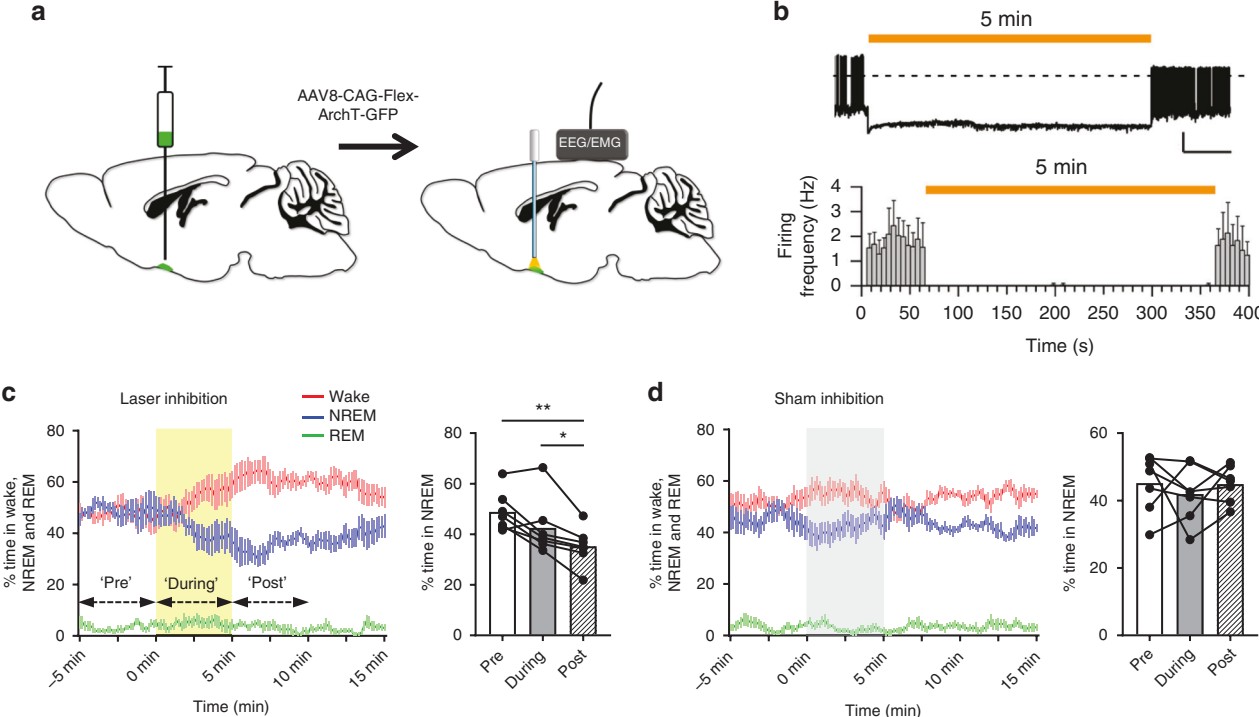

**Fig. 3** Optogenetic inhibition of VLPO$^{GAL}$ neurons decreases NREM sleep. **a** Schematic representation of AAV injections and implantation of optical fibers and EEG/EMG leads. **b** In vitro whole-cell recordings in current clamp mode from VLPO$^{GAL}$ neurons expressing ArchT-GFP. Yellow/orange-light pulses (5 min duration) strongly hyperpolarized (difference in resting membrane potential between the last 10 s prior to laser pulse vs. the last 10 s of laser illumination was $28.4 \pm 3.7$ mV; $p = 0.00026$, paired $t$-test, $n = 7$ neurons in three mice) VLPO$^{GAL}$ neurons expressing ArchT and abolished action potentials (dotted line: 0 mV). **c** In vivo inhibition of VLPO$^{GAL}$ neurons by yellow/orange laser light (593.5 nm wavelength; ~10 mW/mm$^2$ illumination at fiber tip) applied for 5 min every 30 min decreased the percent time spent in NREM sleep during the 12 h dark period. We compared 5 min periods 'before', 'during', and 'after' the photoinhibition using a one-way repeated measures ANOVA for 'treatment groups', followed by Tukey's post hoc test ($F(2, 12) = 15.86$, $p = 0.0005$, $n = 7$ mice. **d** Sham photoinhibition did not alter the amount of NREM sleep in the same mice. Data are Mean ± SEM. $*p < 0.05$, $**p < 0.01$. Scale bars in **b** are 20 mV, 50 s

**Photoinhibition of VLPO$^{GAL}$ neurons decreases NREM sleep.** To study whether VLPO$^{GAL}$ neurons are necessary for sleep, we injected AAV-CAG-Flex-ArchT-GFP ('AAV-ArchT')[33] bilaterally into the VLPO of GAL-IRES-Cre mice ('VLPO$^{GAL}$-ArchT mice') as above (Fig. 3a). AAV-ArchT resulted in specific expression of ArchT-GFP in VLPO$^{GAL}$ neurons, consistent with the distribution of GAL neurons in GAL-IRES-Cre::L10-GFP reporter mice (Supplementary Fig. 1). In the 7 mice used for the experiments, the AAV injections covered an average of 47.2% (range: 37.8–68.5%) of the VLPO.

In vitro, whole-cell current clamp recordings from ArchT-expressing GAL neurons in VLPO slices showed that yellow/orange laser light for 5 min hyperpolarized VLPO$^{GAL}$ neurons and completely prevented action potential firing during the illumination period (Fig. 3b). After the photoinhibition, action potentials resumed at the pre-illumination rate in all neurons (Fig. 3b).

We then measured the sleep-wake effects of photoinhibition of VLPO$^{GAL}$ neurons (5 min continuous light every 30 min for 24 h) (Fig. 3c, d and Supplementary Fig. 4a-d). After ~2 min latency, the percentage time spent in wake gradually climbed from about 47% to about 60% at the end of the 5 min photoinhibition. Wake levels remained in that range for about 5 more min, before gradually returning to baseline. We compared the percentages of sleep-wake states during the 5 min periods immediately prior ('pre'), during ('during'), and after ('post') laser inhibition or sham inhibition. Over the entire 24-h period, the time spent in NREM sleep was decreased during the 5 min post laser inhibition when compared to the pre-laser inhibition ($14.8 \pm 2\%$ reduction;

'pre' vs. 'post': $p = 0.0141$, one-way repeated measures (RM) ANOVA; Supplementary Fig. 4a). The decrease in NREM sleep was more pronounced during the dark period ($27.6 \pm 5\%$ reduction, 'pre' vs. 'post': $p = 0.0027$, one-way RM ANOVA; Fig. 3c) and was accompanied by a proportional increase in wake (Supplementary Table 2). These changes in NREM sleep and wake were mainly due to longer wake bouts (Supplementary Table 2), suggesting that photoinhibition reduced initiation of NREM sleep. REM sleep levels did not differ (Fig. 3c). In contrast to photoinhibition, sham inhibition had no effect on wake, NREM or REM sleep in the same mice (Fig. 3d; Supplementary Fig. 4b,d). These data indicate that acute inhibition of VLPO$^{GAL}$ neurons increases wake by reducing transitions into NREM sleep.

**Chemogenetic activation induces sleep and hypothermia.** Although optogenetics offers many advantages, photostimulation may induce firing patterns that differ from natural patterns. We therefore sought to corroborate our optogenetic findings with chemogenetic experiments, which are also more suitable for long-term stimulation. We bilaterally injected AAV8-hsyn-DIO-hM3Dq-mCherry ('AAV-hM3Dq')[34,35] into the VLPO of GAL-Cre mice ('VLPO$^{GAL}$-hM3Dq mice'; Fig. 4a) and implanted a telemetry transmitter to record EEG, EMG, $T_b$, and locomotion[35]. AAV-hM3Dq injections resulted in specific expression of hM3Dq (as evidenced by mCherry labeling) in POA$^{GAL}$ neurons (Fig. 4b, c). In some mice, the injections were sufficiently laterally placed that they involved neurons only in the VLPO and extended VLPO, avoiding the medial, median, and periventricular preoptic nuclei. We therefore examined the responses in that group

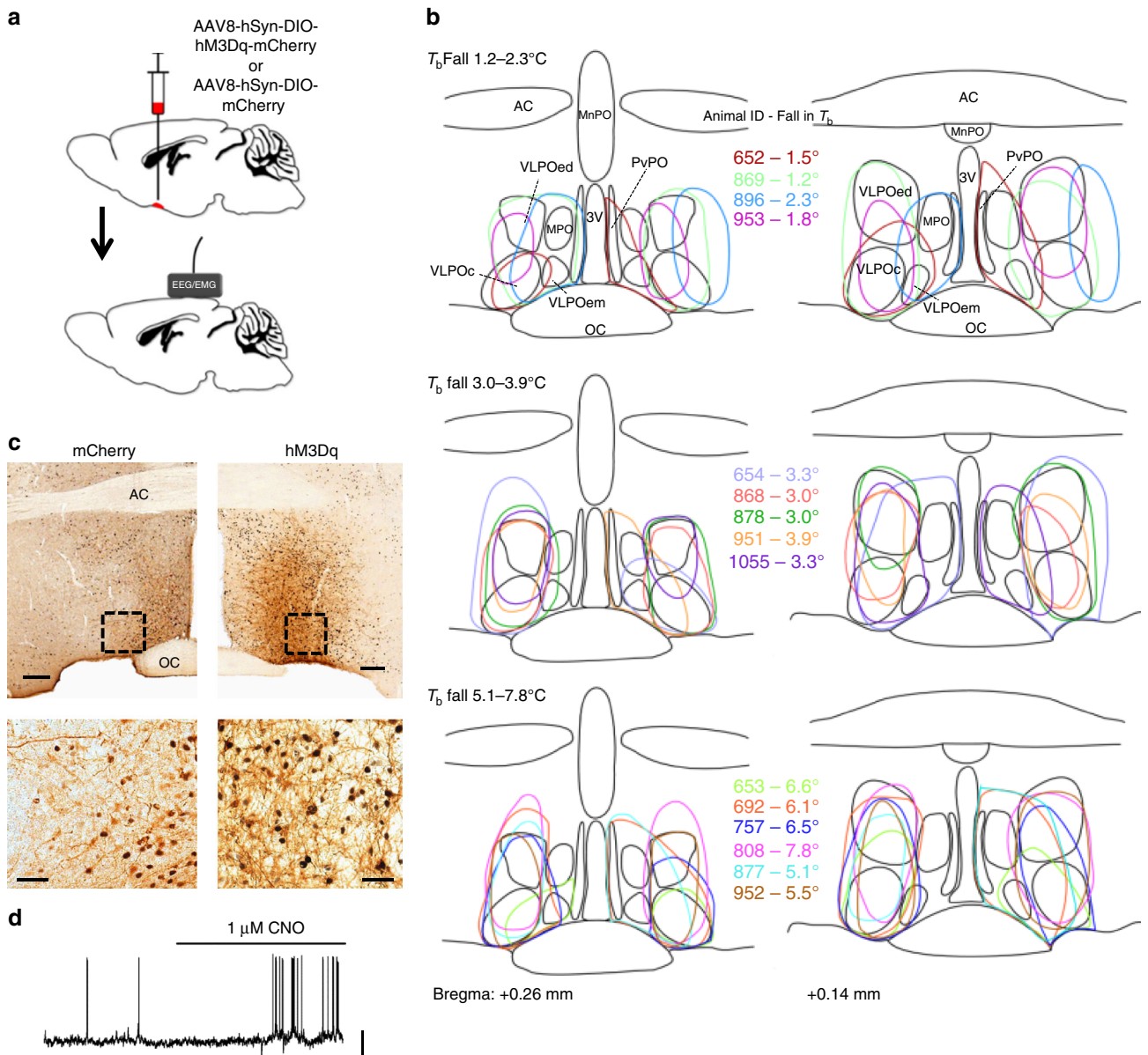

**Fig. 4** Chemoactivation of VLPO$^{GAL}$ neurons by CNO. **a** Schematic representation of viral vector injections. **b** Outlines of AAV-hM3Dq injection sites in VLPO at two levels (equivalent to AP: +0.26 and AP: +0.14 in Ref. [70]; Animal IDs ($n = 15$ mice) and the magnitude of hypothermia observed after chemoactivation in each mouse is represented in matching colors **c** cFos expression (black nuclei) in mCherry-expressing neurons (brown) 3 h after i.p. injections of CNO (0.3 mg/kg) in VLPO$^{GAL}$-mCherry mice (left) and VLPO$^{GAL}$-hM3Dq mice (right). **d** Representative trace of in vitro activation of an hM3Dq-expressing VLPO$^{GAL}$ neuron with 1 μM CNO ($n = 5$ cells from five mice). 3 V, 3rd Ventricle; AC Anterior commissure; OC optic chiasm; MnPO Median preoptic nucleus; MPO Medial preoptic area; PvPO Periventricular preoptic area; VLPO Ventrolateral preoptic area; VLPOc. VLPO core; VLPOed, VLPO extended dorsal; VLPOem, VLPO extended medial. Scale bars in **c** 200 μm (upper panels) and 50 μm (lower panels) and **d** 20 mV, 60 s

separately, to see whether they differed from the cases where the more medial, small celled preoptic GAL-expressing neurons were also involved. Injection of AAV-hM3Dq in WT littermates (not expressing Cre) resulted in no mCherry expression. We also used GAL-IRES-Cre mice injected with AAV-DIO-mCherry in the VLPO (VLPO$^{GAL}$-mCherry mice) as negative controls.

In whole cell, current clamp recordings in vitro, bath application of CNO depolarized ($10.5 \pm 2.9$ mV; $-52.9 \pm 3.2$ mV post-CNO vs. $-63.4 \pm 4.4$ mV pre-CNO; $p = 0.089$, paired $t$-test) hM3Dq-expressing VLPO$^{GAL}$ neurons and increased firing rates ($0.7 \pm 0.1$ impulses/s post CNO vs. 0.03 impulses/s pre-CNO; $p = 0.001$, paired $t$-test; Fig. 4d). In vivo, CNO (0.3 mg/kg i.p.) increased cFos expression (Fig. 4c) in hM3Dq-expressing neurons

in the VLPO, indicating that CNO activated hM3Dq-expressing GAL neurons.

To assess changes in sleep-wake behavior and $T_b$ with chemoactivation of VLPO$^{GAL}$ neurons, we injected CNO (0.3 mg/kg, i.p.) or saline (vehicle) at the onset of the dark period and early in the light period. Administration of CNO at dark onset (19:00 h) increased NREM sleep 71% over the next 4 h when compared to saline, while CNO injection at 10:00 h increased NREM sleep 21% during the first 12 h (three bins of 4 h each) (Fig. 5a, b and Supplementary Fig. 5). This increase in NREM sleep was due to a substantial increase in the number (but not the duration) of NREM sleep bouts, further demonstrating that VLPO$^{GAL}$ neurons promote NREM sleep by increasing wake-to-

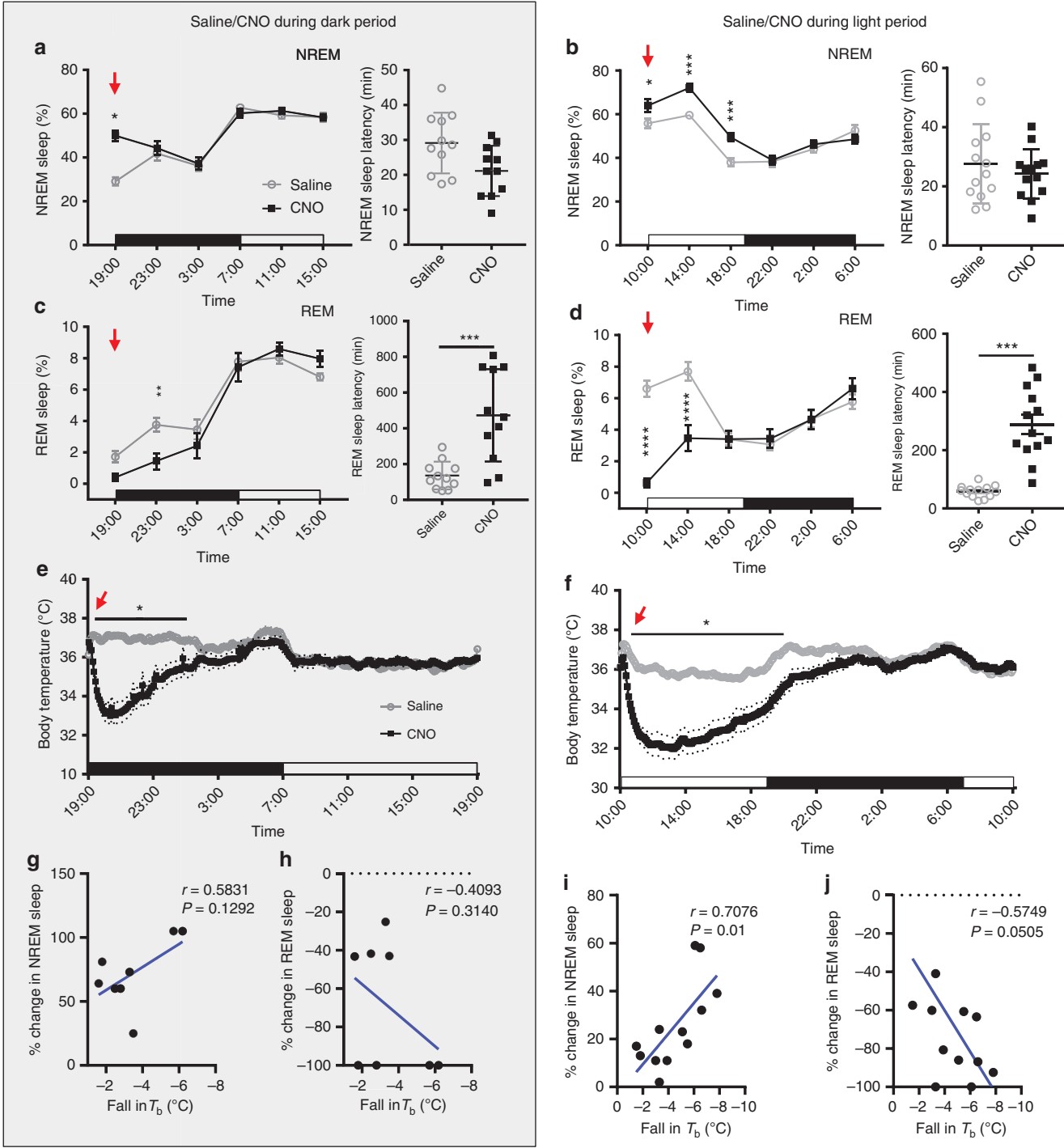

NREM transitions (Supplementary Fig. 5e-h,). Unlike photostimulation, which entrained the EEG to the frequency of the laser bursts, CNO did not alter the EEG spectrum during NREM sleep (Supplementary Fig. 5c,d). In contrast to the increase in NREM sleep, REM sleep was suppressed by 66% and 71% during the first 8 h post CNO in the dark and light periods, respectively, and this REM sleep suppression was accompanied by longer latencies to enter REM sleep (Fig. 5c, d). None of these sleep-wake changes were observed in control GAL-IRES-Cre mice that received an AAV encoding only mCherry (Supplementary Fig. 6).

In addition to sleep-wake changes, chemoactivation of hM3Dq-expressing VLPO$^{GAL}$ neurons markedly reduced $T_b$ during both the light and dark periods. $T_b$ began to drop 5–10

min after CNO injections, fell ~4–6°C within ~2 h, and then returned to baseline levels after 12–14 h (Fig. 5e, f). As this hypothermia was not observed with photostimulation which predominantly activated GAL neurons in the lateral POA (optical fiber tips were mainly dorsal to the VLPO cluster; Supplementary Fig. 2j), we examined whether differences in the injection sites in optogenetic and chemogenetic experiments could contribute to this hypothermic response. Examination of the injection sites that caused the greatest fall in temperature showed that many of them were confined to the core and dorsal extended VLPO, with little or no involvement of the medial, median, or periventricular preoptic GAL neurons (e.g., cases 757, 808). As expected, these cases were also among those showing the greatest increase in

**Fig. 5** Chemoactivation of VLPO$^{GAL}$ neurons promotes NREM sleep and hypothermia. **a**, **b** Percentages of NREM sleep every 4 h for 24 h after saline or CNO administration (red arrow) at 19:00 h **a** or 10:00 h **b** and respective NREM sleep latencies. Two-way repeated measures (RM) ANOVA for the first 12 h after treatment for 'time' and 'compound injected', followed by Sidak's post hoc test (19:00 h injections ($n = 9$ mice): time $F_{(2,32)} = 2.398$, $p = 0.1070$, compound injected $F_{(1,16)} = 22.76$, $p = 0.0002$; 10:00 h injections ($n = 13$ mice): time $F_{(2,48)} = 77.29$, $p < 0.0001$, compound injected $F_{(1,24)} = 28.92$, $p < 0.0001$). **c**, **d** Percentages of REM sleep after saline or CNO at 19:00 h **c** or 10:00 h **d** with respective REM sleep latencies. Two-way RM ANOVA followed by Sidak's post hoc test (19:00 h injections ($n = 9$ mice): time $F_{(2,32)} = 10.77$, $p = 0.0003$, compound injected $F_{(1,16)} = 7.763$, $p = 0.0132$; 10:00 h injections ($n = 13$ mice): time $F_{(2,48)} = 11.75$, $p < 0.0001$, compound injected $F_{(1,24)} = 41.06$, $p < 0.0001$); Wilcoxon matched-pairs signed-rank test for REM sleep latencies (19:00 h injections $p = 0.0010$; 10:00 h injections $p = 0.0002$). **e**, **f** $T_b$ after saline or CNO at 19:00 h **e** or 10:00 h **f**; dotted lines represent SEM; Two-way RM ANOVA for the first 12 h after treatment for 'time' and 'compound injected', followed by Sidak's post hoc test (19:00 h injections: time $F_{(143, 4004)} = 19.4$, $p < 0.0001$, compound injected $F_{(1,28)} = 19.09$, $p = 0.0002$; $n = 15$ mice; 10:00 h injections: time $F_{(143,4290)} = 65$, $p < 0.0001$, compound injected $F_{(1,30)} = 29.78$, $p < 0.0001$; $n = 16$ mice). **g**, **i** Correlation between percentage change in NREM sleep and fall in $T_b$ after CNO at 19:00 h **g** or 10:00 h **i**; Pearson correlation (19:00 h injections: $r = 0.5831$, $p = 0.1292$, $n = 8$ mice; 10:00 h injections: $r = 0.7076$, $p = 0.01$; $n = 12$ mice). **h**, **j** Correlation between percentage change in REM sleep and fall in $T_b$ after CNO at 19:00 h **h** or 10:00 **j**; Pearson correlation (19:00 h injections $r = 0.4093$, $p = 0.3140$, $n = 8$ mice; 10:00 injections: $r = 0.5749$, $p = 0.0505$; $n = 12$ mice). Red arrows indicate saline/CNO injections. Data are Mean ± SEM, except sleep latencies which are Mean ± SD, *$p < 0.05$, **$p < 0.01$, ***$p < 0.001$, ****$p < 0.0001$

NREM sleep after CNO administration. The percentage increase in NREM sleep positively correlated with the magnitude of fall in $T_b$ (Fig. 5g, i). Conversely, the percentage change in REM sleep negatively correlated with the magnitude of hypothermia (Fig. 5h, j). Although we did not test whether the same individual neurons cause both the fall in $T_b$ and changes in sleep, it appears that both responses are caused by activation of a single population of VLPO$^{GAL}$ neurons.

We were also concerned, in light of recent reports that CNO may be converted to clozapine in vivo[36], whether CNO injection could cause hypothermia by itself, but it failed to do so in GAL-IRES-Cre mice injected with AAV-mCherry (Supplementary Fig. 6j,l). It is therefore more likely that the profound hypothermia is due to differences in firing rate/pattern of VLPO$^{GAL}$ neurons with chemoactivation (vs. photoactivation) that caused different downstream responses such as release of other neurotransmitters/peptides.

**Chemoactivation of VLPO$^{GAL}$ neurons at different ambient temperatures.** Sleep and thermoregulation are closely intertwined in many ways[37–40]. For example, $T_b$ normally falls about 1 °C during sleep in mice[41]. On the other hand, severe hypothermia may promote arousal, fragment NREM sleep, reduce EEG delta power during NREM sleep, and suppress REM sleep[37–39,42]. We therefore hypothesized that chemoactivation of VLPO$^{GAL}$ neurons would more strongly increase the quantity and quality of sleep (more consolidated sleep bouts, increased EEG delta power during NREM, and more REM sleep) if the mice were provided with 'thermal comfort' by exposure to an ambient temperature ($T_a$) that would blunt the fall in $T_b$. To test this hypothesis, we chemogenetically activated VLPO$^{GAL}$ neurons in mice housed at thermoneutral (29 °C) or warm $T_a$ (36 °C, empirically determined to prevent the fall in $T_b$, Fig. 6a).

As predicted, CNO injections produced larger increases in NREM sleep at both 29 °C and 36 °C compared to 22 °C or injections of saline (Fig. 6c; Supplementary Fig. 7). Surprisingly, a $T_a$ of 29 °C, which is generally thought to be near thermoneutral for mice, did not prevent the hypothermia due to CNO, and only at a $T_a$ of 36 °C was the CNO-induced hypothermia completely prevented (Fig. 6b). The amount of NREM sleep after CNO at $T_a$ 36 °C was substantially greater and the NREM sleep latency was shorter than with CNO injections at $T_a$ 22 °C (Fig. 6c, d) while the mice at $T_a$ of 29 °C showed intermediate responses. CNO also increased NREM EEG delta power at $T_a$ 36 °C (Fig. 6e) when compared to CNO at $T_a$ 29 °C, suggesting deeper sleep at higher $T_a$.

In contrast, CNO given at dark onset suppressed REM sleep at all $T_a$ (Fig. 6f). Whether this was due to activation of VLPO$^{GAL}$

neurons is difficult to determine as REM sleep is generally low during the dark period, and high $T_a$ alone can suppress REM sleep[43]. To address these issues, we administered CNO or saline to VLPO$^{GAL}$-hM3Dq mice at 29 °C during the light period when REM sleep is more abundant. We found that CNO decreased REM sleep during the first 4 hours at 29 °C, but REM sleep rose quickly during the next 12 h until it was fully recovered. Cumulative REM sleep amounts in the 16 h post CNO were higher than at $T_a$ 22 °C and were comparable to post-saline data at $T_a$ 22 °C or $T_a$ 29 °C (Fig. 6g). These results indicate that a $T_a$ of 29 °C, which does not prevent hypothermia or initial REM sleep loss, can rescue REM sleep amounts by enhancing REM sleep rebound after hypothermia.

**Activation of POA$^{GAL}$ neurons attenuates sleep-onset insomnia.** To explore the translational relevance of our findings for insomnia patients, we examined whether chemoactivation of VLPO$^{GAL}$ neurons can improve sleep in an animal model of insomnia. Acute exposure to a novel environment (e.g. transfer to the new cage) reduces sleep in mice for 3–4 h, akin to sleep-onset insomnia in humans[44]. Therefore, we administered saline or CNO to VLPO$^{GAL}$-hM3Dq mice at 10:00 h (3 h into the light period; at 22 °C) and immediately transferred them to a new cage (similar to home cage) with fresh bedding and nesting material (Fig. 7a). CNO injections increased NREM sleep about threefold during the first 4 h compared to saline treatment (Fig. 7b and Supplementary Fig. 8c) and also significantly shortened NREM sleep latencies (Fig. 7c and Supplementary Fig. 8d). In addition, the hyperthermia usually associated with exposure to a new cage was attenuated by CNO (Fig. 7d and Supplementary Fig. 8g). On the other hand, the hypothermia observed after CNO treatment in the home cage was still present, albeit with a lower magnitude. These findings indicate that activation of VLPO$^{GAL}$ neurons can attenuate sleep-onset insomnia and hyperthermia induced by the stress of a novel environment.

**Discussion**
Although, the role of the POA in sleep has been recognized for decades, the neurochemical identity of sleep-promoting neurons has remained controversial. Based on cFos expression and projections to the TMN[9–11], GAL neurons in the VLPO (including the extended VLPO) have been suspected to play a critical role in the regulation of NREM sleep. Although lesions of the VLPO in rats dramatically reduced sleep time[12–14], and loss-of-VLPO$^{GAL}$ neurons in humans with age is associated with fragmentation and loss-of-consolidated sleep[15], it remained unclear if VLPO$^{GAL}$ neurons actually promote sleep[16]. We find that selective

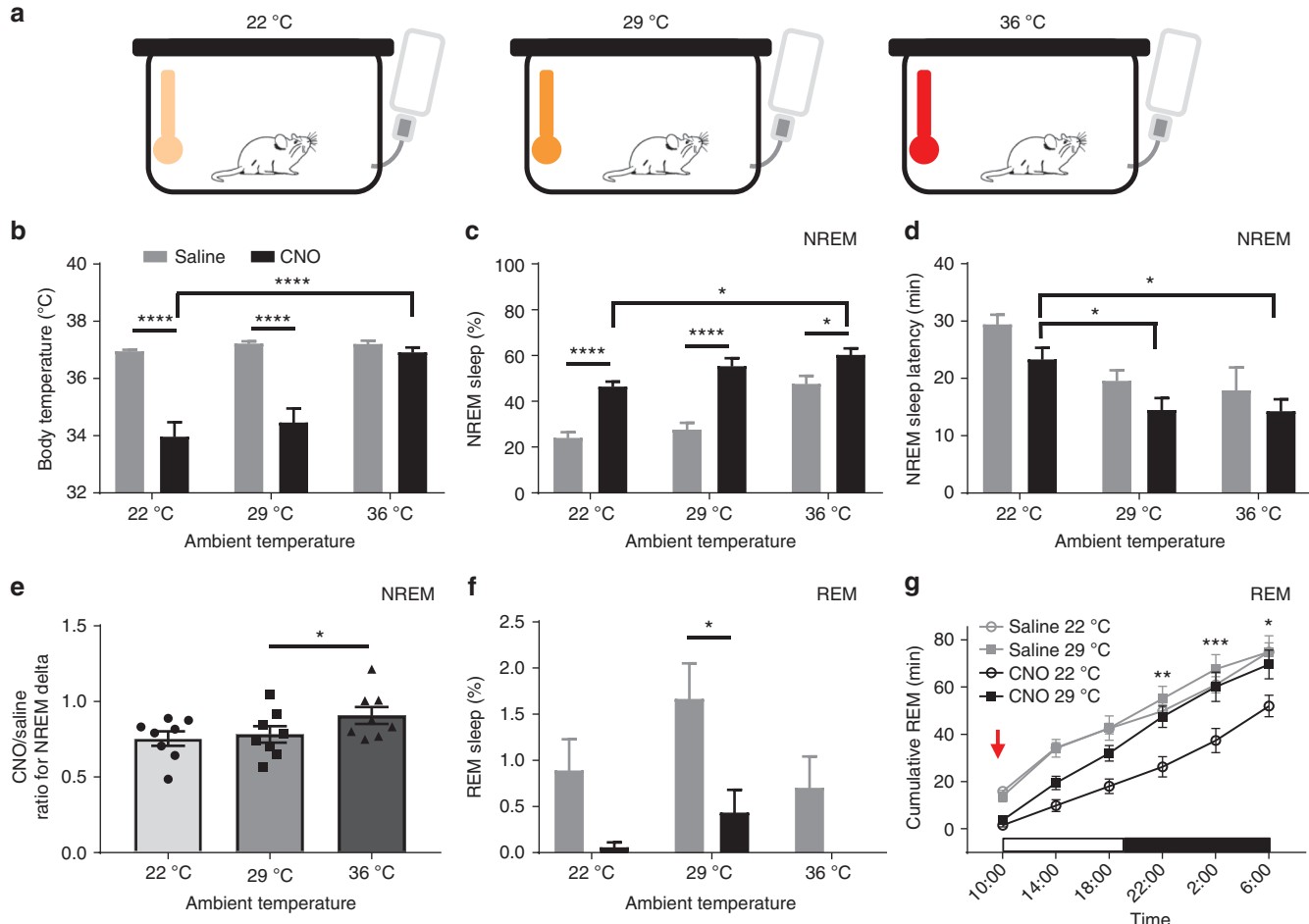

**Fig. 6** Warm ambient temperatures ($T_a$) enhance NREM sleep with chemoactivation of VLPO[GAL] neurons. **a** Schematic of experimental paradigm: baseline $T_a$ 22 °C and exposure to $T_a$ 29 °C and 36 °C. **b** Mean $T_b$ for 2 h after saline/CNO (0.3 mg/kg) injections at 19:00 h in different $T_a$: Two-way repeated measures (RM) ANOVA for 2 h after treatment for '$T_a$' and 'compound injected', followed by Sidak's post hoc test ($T_a$: $F_{(2,28)} = 46.87$, $p < 0.0001$, compound injected: $F_{(1,14)} = 26.26$, $p = 0.0002$; $n = 8$ mice). **c** Percentages of NREM sleep for 2 h after saline or CNO treatment in different $T_a$: Two-way RM ANOVA followed by Sidak's post hoc test ($T_a$: $F_{(2,28)} = 28.29$, $p < 0.0001$, compound injected: $F_{(1,14)} = 40.08$, $p < 0.0001$; $n = 8$ mice). **d** NREM latencies after saline or CNO injections in different $T_a$: Two-way RM ANOVA followed by Sidak's post hoc test ($T_a$: $F_{(2,28)} = 0.1369$, $p = 0.0002$, compound injected: $F_{(1,14)} = 5.12$, $p = 0.0401$; $n = 8$ mice). **e** CNO/Saline ratio of NREM delta power for 2 h after saline or CNO treatment in different $T_a$. One-way ANOVA (Friedman test) for 2 h after treatment for '$T_a$', followed by Dunn's multiple comparisons test ($p = 0.0303$); **f** Percentages of REM sleep for 2 h after saline or CNO treatment in different $T_a$: Two-way RM ANOVA followed by Sidak's post hoc test ($T_a$: $F_{(2,28)} = 3.228$, $p = 0.0548$, compound injected: $F_{(1,14)} = 20.19$, $p = 0.0005$; $n = 8$ mice). **g** Cumulative REM sleep amounts every 4 h for 24 h after saline or CNO at $T_a$ 29 °C during the light period (10:00 h). Note that animals at 29 °C who received CNO had less REM sleep in the initial 4 hours, but by the end of the next 16 hours REM rebound resulted in catching up to animals treated with saline: Two-way RM ANOVA for 'time' and 'compound injected', followed by Sidak's post hoc test (time: $F_{(5,170)} = 391.5$, $p < 0.0001$, compound injected: $F_{(3,34)} = 12.79$, $p < 0.0001$; $n = 13$ mice ($T_a$ 22 °C), $n = 6$ mice ($T_a$ 29 °C). Data are Mean ± SEM, *$p < 0.05$, **$p < 0.01$, ***$p < 0.001$, ****$p < 0.0001$.

activation of VLPO[GAL] neurons increases NREM sleep and their inhibition decreases NREM sleep, thereby establishing a pivotal role for VLPO[GAL] neurons in sleep promotion. NREM sleep with photoactivation of VLPO[GAL] neurons resembled normal sleep behaviorally; mice reduced locomotor activity in preparation for sleep, assumed a sleep posture within their nest and transitioned into sleep. In addition, chemoactivation of VLPO[GAL] neurons produced severe hypothermia, highlighting the close connection between NREM sleep and heat loss. Although our experiments did not address whether individual VLPO[GAL] neurons contributed to both, the neurons contributing to these functions are at least intermixed, as different injections involved different proportions of the VLPO core and the dorsal and medial extended VLPO[GAL] neurons, but the amount of sleep and hypothermia correlated across the entire set of experiments.

At first glance, our results may appear to contradict a recent report by Chung et al. that 10 Hz optogenetic activation of VLPO[GAL] neurons causes wakefulness, rather than sleep[16]. However, VLPO sleep-active neurons generally fire no faster than 2–5 Hz, even during deep NREM sleep[16,25,32]. We find that VLPO[GAL] neurons fire reliably with low frequency photo-stimulation (0.5–4 Hz) in vitro, and in vivo photostimulation at these frequencies clearly increases NREM sleep. In contrast, higher frequency stimulation (8 or 16 Hz) causes depolarization block in vitro, and in vivo these frequencies either decrease or have no effect on NREM sleep. We suspect that the wakefulness with 10 Hz photostimulation in the prior study[16] was caused by conduction block and acute silencing of VLPO[GAL] neurons.

Interestingly, Chung et al. also photostimulated POA neurons at 10 Hz that had transported ChR2-AAV retrogradely from the TMN, and found that they promote sleep[16]. Most of these are

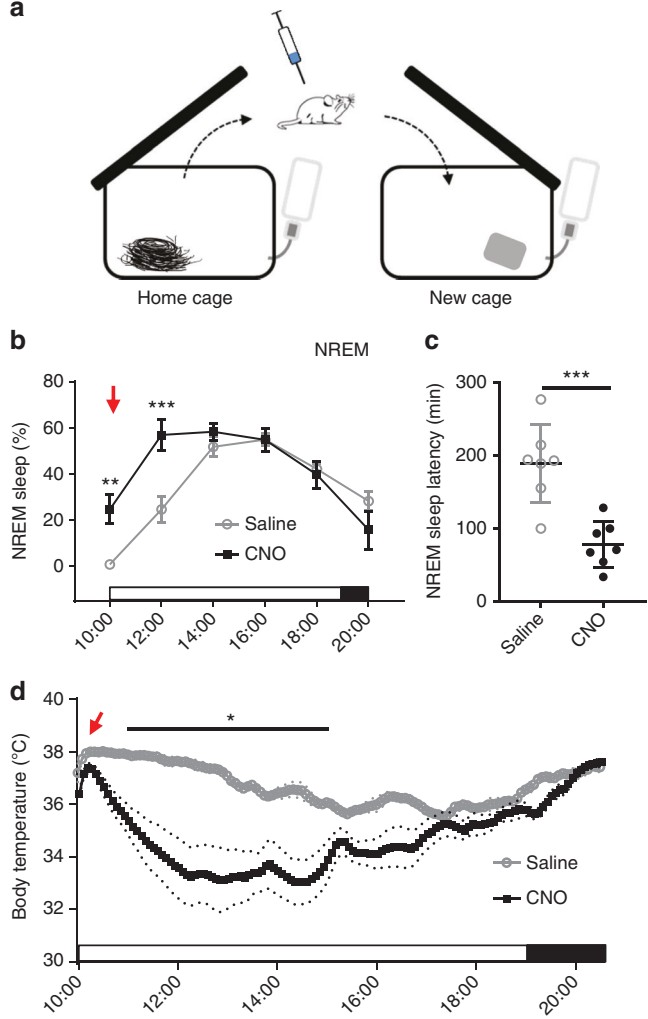

**Fig. 7** Chemogenetic activation of VLPO[GAL] neurons attenuates sleep-onset insomnia. **a** Schematic of experimental setup showing mice introduced to a new cage with fresh nesting material immediately after i.p. injections of either saline or CNO. **b** Percentages of NREM sleep every 2 h for 12 h after saline or CNO (0.3 mg/kg) injections and cage change at 10:00 h. Typically, mice introduced to a new cage during the day (when they usually sleep about 60% of the time) have reduced sleep for 2–4 h with a prolonged latency to first NREM sleep. This is rescued by activation of VLPO[GAL] neurons. Two-way repeated measures (RM) ANOVA for the first 6 h after treatment for 'time' and 'compound injected', followed by Sidak's post hoc test (time: $F_{(3,36)} = 64.72$, $p < 0.0001$, compound injected: $F_{(1,12)} = 8.644$, $p = 0.0124$; $n = 7$ mice). **c** NREM sleep latencies after injections and cage change: Wilcoxon matched-pairs signed-rank test ($p = 0.0156$; $n = 7$ mice). **d** $T_b$ after injections and cage change. Two-way RM ANOVA for the first 6 h after treatment for 'time' and 'compound injected', followed by Sidak's post hoc test (time: $F_{(71,852)} = 14.72$, $p < 0.0001$, compound injected: $F_{(1,12)} = 15.21$, $p = 0.0021$; $n = 7$ mice). Red arrows indicate saline/CNO injections. The data are Mean ± SEM, except sleep latencies which are Mean ± SD, *$p < 0.05$, **$p < 0.01$, ***$p < 0.001$

likely to have been VLPO[GAL] neurons, as about 85% of preoptic neurons retrogradely labeled from the TMN in rats contain GAL[10]. However, the remaining 15% leaves room for additional neurons projecting to the TMN that do not express GAL. Consistent with this idea, most VLPO neurons projecting to the TMN fired in the range of 2–6 Hz during sleep and 1 Hz during wake[16].

However, other TMN-projecting POA neurons fired at rates of up to 12 Hz, especially during REM sleep. These faster firing neurons may produce neurotransmitters other than GAL and may be activated when photostimulated at 10 Hz, but they would not have been included in our study, which was limited to VLPO[GAL] neurons.

As VLPO[GAL] neurons contain GABA and other neuropeptides besides GAL[10], it is possible that sleep and $T_b$ changes induced by activation of VLPO[GAL] neurons may be mediated by neurotransmitters or neuropeptides other than GAL. General photoactivation of POA GABAergic neurons produced only a minor increase in sleep[45] or even wakefulness[16], although there are other GABAergic neurons in the VLPO region that do not contain GAL[16,46]. On the other hand, photoactivation of only GABAergic POA neurons that project to the TMN (85% of which contain GAL) produced sleep[16]. These experiments suggest that there are several populations of GABAergic neurons in and around the POA (e.g., the basal forebrain GABAergic arousal neurons are just lateral to the VLPO core), some of which may promote sleep while others promote wake[25,32]. Further investigation is needed to identify the specific neurotransmitter in GAL neurons involved in these functions.

An important question is whether the VLPO[GAL] neurons help to initiate or to maintain NREM sleep. Neurotoxic lesions of the VLPO severely shortened NREM sleep bouts[12,13], but animals transitioned into NREM sleep at a normal frequency[13]. This was interpreted as the VLPO being critical for NREM sleep maintenance rather than initiation. However, in those chronic experiments, animals likely had an abnormally high sleep drive due to accumulated sleep loss, and this high sleep pressure may have increased transitions into NREM sleep. We found that VLPO[GAL] photoactivation and chemoactivation increased transitions into NREM sleep but did not lengthen NREM sleep bouts. Similarly, chemoactivation reduced NREM sleep latency in an insomnia model where the drive for wake in mice is extremely high. These findings suggest a critical role for VLPO[GAL] neurons in the initiation of NREM sleep.

Photostimulation of VLPO[GAL] neurons did not alter REM sleep whereas chemoactivation strongly suppressed this state. While photostimulation drives neuronal activity rhythmically, chemoactivation likely makes neurons more responsive to natural inputs, and this could drive different downstream responses that impact REM sleep. Alternatively, hypothermia and thermal stress during chemogenetic activation of VLPO[GAL] neurons at room temperature may also have suppressed REM sleep because (a) REM sleep loss with chemoactivation of VLPO[GAL] neurons positively correlated with the degree of hypothermia, and (b) REM sleep was recovered with exposure to a thermoneutral $T_a$ (29 °C). This is unsurprising as changes in $T_a$ and $T_b$ disrupt REM sleep much more than NREM sleep[37,39,43,47–50] and REM sleep is maximal at a thermoneutral $T_a$ (27–32 °C in mice) and decreases with $T_a$ above or below these levels[51–53].

The severity of hypothermia with chemoactivation of VLPO[GAL] neurons demonstrates an important role for these neurons in thermoregulation in addition to sleep-wake regulation. In our previous work in rats[10], we found that VLPO neurons project to the raphe pallidus nucleus, an area that mediates hyperthermic responses, and that most of the neurons contributing to this projection are in the region of the dorsal extended VLPO[20]. McAllen and colleagues found that neurons in this region, which they called the caudolateral POA to distinguish it from the median preoptic nucleus (their "rostromedial" POA), promote heat loss via tail artery vasodilation[54,55]. Currently, it is unknown whether the neurons contributing to this pathway are specifically galaninergic, but our results suggest that at least some of them are.

NREM sleep is usually associated with a small decrease in $T_b$ (~1 °C) and wakefulness is associated with an increase. In contrast, higher $T_b$ can promote NREM sleep (e.g. low grade fever) and hypothermia causes arousal[56–59]. However, in our chemogenetic experiments, we observed a substantial increase in NREM sleep despite dramatic hypothermia, indicating that both responses were consequences of VLPO$^{GAL}$ activation and that increased NREM sleep was not secondary to the hypothermia. The additional increase in NREM amounts and decrease in sleep latency observed when activating VLPO$^{GAL}$ neurons in a warm environment suggests that hypothermia (and the associated thermal discomfort) caused some arousal effects at room temperature.

Thermal and sleep changes induced by VLPO$^{GAL}$ activation highlight the strong link between these two functions as most sleep-active POA cells in rats are also warm-sensitive and their activation may drive thermoregulatory responses[24,25,51,57]. Local preoptic warming (which activates warm-sensitive neurons) can promote NREM sleep and reduce $T_b$, whereas local cooling does the opposite[8,51,56]. Moreover, chemoactivation of POA neurons that expressed cFos during recovery sleep, resulted in similarly profound hypothermia[60] suggesting that sleep and thermoregulatory mechanisms are strongly coupled in the same neurons in the POA. Our present results indicate that VLPO$^{GAL}$ neurons play a key role in this coupling.

The anatomical projections of VLPO$^{GAL}$ neurons suggest pathways through which they can promote NREM sleep and heat loss. Conditional tracing showed that VLPO$^{GAL}$ neurons in mice project to targets similar to VLPO neurons in rats[10]. Both densely innervate wake-promoting regions, including the LH, TMN, and LC and moderately innervate the lateral PB and the PPT. Thus, NREM sleep-promotion by VLPO$^{GAL}$ neurons may be mediated by simultaneous inhibition of multiple-wake promoting regions. On the other hand, VLPO$^{GAL}$ stimulation may cause hypothermia through projections to the DHA, DMH, lateral PAG, RPa and PPR, which are considered critical nodes for heat production and retention. Inhibition of these structures by VLPO$^{GAL}$ neurons may reduce $T_b$ by suppressing thermogenesis and promoting vasodilation.

Collectively, our optogenetic studies demonstrate that activation of VLPO$^{GAL}$ neurons within their typical physiological firing range causes sleep. Additionally, chemoactivation of VLPO$^{GAL}$ neurons also produced a drastic fall in $T_b$. However, the relative degree to which VLPO$^{GAL}$ neurons contribute to sleep vs. thermoregulation may depend on their firing patterns, projections and environmental conditions.

Finally, the deep hypothermia seen with chemoactivation of VLPO$^{GAL}$ neurons, which is far beyond that seen with normal wake-sleep cycles, suggests a role in torpor. Torpor is a deeply hypothermic and unresponsive state, with a slow wave EEG, seen in mammals (from mice to bears) when food is too scarce for homeothermy[61,62]. Torpor permits low metabolism, which can sustain life when food is unavailable. The relationship between torpor and sleep has been debated[63–65], but our observations suggest that prolonged activation of VLPO neurons, as with chemoactivation, may push VLPO$^{GAL}$ neurons into a firing state that fosters torpor. Thus, rather than just serving as a wake-sleep switch, the VLPO$^{GAL}$ neurons may serve as a switch with three positions: wake, sleep, and torpor.

## Methods

**Mice.** We used male GAL-IRES-cre transgenic mice on a C57BL/6J background (generated by Dr. Münzberg)[28,66] and their wild-type (WT) littermates (aged 8–12 weeks at the start of the experiments). Eutopic expression of Cre was verified by immunostaining for the GAL peptide[28] and by crossing GAL-Cre mice with a reporter line expressing green fluorescent protein (GFP) and comparing the expression pattern to that of *Gal* mRNA in the Allen Brain Atlas[28,66]. Mice were maintained under standard vivarium conditions (12 h : 12 h light-dark cycle with lights on at 07:00 h; 22 ± 1 °C ambient temperature). Care of the animals met National Institutes of Health standards, as set forth in the Guide for the Care and Use of Laboratory Animals, and all protocols were approved by the BIDMC Institutional Animal Care and Use Committee.

**Viral vectors.** We obtained Cre-dependent adeno-associated viral vectors for optogenetic stimulation (AAV8-EF1α-DIO-ChR2(H134R)-mCherry), optogenetic inhibition (AAV8-CAG-Flex-ArchT-GFP), chemogenetic stimulation (AAV8-hSyn-DIO-hM3Dq-mCherry) as well as control viral vectors (AAV8-EF1α-DIO-mCherry) from the Vector Core at the University of North Carolina, USA. All viral vectors had titer concentrations of $3-6 \times 10^{12}$ vector genomes per milliliter. The specificity of these viral vectors has been confirmed in various Cre lines including GAL-IRES-Cre mice used in this study[35,66]. In our hands, we observed that 88.6% of ChR2-expressing neurons were positive for *Gal* mRNA (by in situ hybridization using RNAScope), as well as 88.5% of ArchT-expressing neurons and 90.0% of hM3Dq-expressing neurons.

**Anterograde tracing.** For tracing anterograde projections of VLPO$^{GAL}$ neurons, we anesthetized GAL-IRES-Cre mice ($n = 8$) with ketamine/xylazine (100 and 10 mg/kg, respectively, i.p.) and injected[67] them with 18 nl of AAV8-EF1α-DIO-ChR2 (H134R)-mCherry unilaterally in the VLPO (anteroposterior: +0.15 mm from bregma, ventral: 5 mm below the dura, lateral +0.5 mm). Six weeks after the injections, we transcardially perfused the mice under deep anesthesia, with phosphate-buffered saline (PBS) followed by 10% formalin (Fisher Scientific). We post-fixed brains in formalin overnight and stored them in 20% sucrose (for cryoprotection). We then cut the brains into 3 series of 40 µm sections on a freezing microtome and immunostained one series for mCherry to map the anterograde projections of mCherry-expressing VLPO$^{GAL}$ neurons.

**Surgery.** GAL-IRES-Cre mice were anesthetized (100 mg/kg ketamine +10 mg/kg xylazine; i.p.) and placed in a stereotaxic frame (David Kopf Instruments). Mice for optogenetic experiments received microinjections of 36 nl of AAV [AAV8-EF1α-DIO-ChR2(H134R)-mCherry ($n = 10$ mice), AAV8-CAG-Flex-ArchT-GFP ($n = 7$ mice) or AAV8-EF1α-DIO-mCherry ($n = 6$ mice)] into the VLPO (anteroposterior: +0.15 mm from bregma, ventral: 5 mm below the dura, lateral ±0.5 mm from the midline) bilaterally. Mice for chemogenetic experiments received the same volume of AAV8-hsyn-DIO-hM3Dq-mCherry ($n = 13$ mice) or AAV8-hsyn-DIO-mCherry ($n = 8$ mice) into the VLPO. After the injections, mice for optogenetic experiments were implanted with bilateral optical fibers 0.2 mm above the VLPO[27] and electrodes for recording electroencephalogram (EEG), electromyogram (EMG)[30]. A miniature telemetry transmitter for recording $T_b$ and locomotor activity (LMA) (TA10-TAF20; Data Sciences International., USA)[31] was also implanted into mice for the optogenetic stimulation experiments. Finally, mice for chemogenetic experiments were implanted with miniature telemetry transmitters (TLM2-F20EET; Data Science International, USA) for recording EEG, EMG, $T_b$, and LMA[35].

**Optogenetic stimulation and inhibition.** Mice were connected to the EEG/EMG wires and optical patch cords >2 weeks prior to recordings to facilitate proper habituation and natural sleep patterns. For photoactivation experiments, we used a function generator (PCGU1000, Velleman Instruments, USA) to stimulate both, VLPO$^{GAL}$-ChR2 mice ($n = 10$) and VLPO$^{GAL}$-mCherry mice ($n = 6$) with 0.5, 1, 2, 4, 8, and 16 Hz light pulses (10 ms per pulse, 10 mW light output at the fiber tip; the order of stimulation frequencies was randomized) for 2 h (21:00–23:00 h) and recorded EEG/EMG, video, body temperature, and locomotion data for 2 h prior to, during and 2 h after stimulation time. For photoinhibition experiments, we applied continuous pulses of orange/yellow light for 5 min every 30 min for 24 h and recorded EEG/EMG signals and video. In the sham inhibition condition the laser was not switched on, but the function generator still produced 5 min pulses every 30 min. EEG and EMG signals were amplified (A.M systems, USA), digitized and recorded using vital recorder software (Kissei Comtek, Japan)[13]. $T_b$ and LMA were recorded telemetrically using Dataquest A.R.T 4.1. software (Data Sciences International., USA)[31,35]. The order of stimulation frequencies for the photoactivation experiments was counterbalanced between animals and >3 days of rest were allowed between stimulation sessions to avoid habituation to induced sleep patterns. To prevent light leakage into the cage, we used shielded optical patch cords to connect the implanted optical fibers (200 µm diameter, ThorLabs) to the DPSSL laser diodes (473 nm, Laserglow, Canada) and employed heat-shrink tubing as well as black nail polish to further shield all optical connections and even the dental cement on the implant itself[68].

**Chemogenetic stimulations.** Four weeks after the surgery, VLPO$^{GAL}$-hM3Dq mice and VLPO$^{GAL}$-mCherry mice were injected with either vehicle (saline) or CNO (0.3 mg/kg; Sigma, USA) and telemetric recordings of sleep-wake behavior, $T_b$ and LMA were performed for 24 h after each injection using Dataquest ART 4.1 software (Data Sciences International, USA)[35]. Each animal received injections of saline and CNO under different conditions: at 10:00 h (light period; ZT 3), 19:00

h (dark period; ZT 12), novel cage exposure, 22 °C, 29 °C, and 36 °C temperatures. For the novel cage experiments, mice were transferred immediately after i.p. injections to another cage (similar to the home cage to which the mice were habituated for at least a week) with new bedding and nesting material. These injections and cage transfer were conducted at 10:00 h. Saline and CNO injections and experimental conditions were randomized and there was at least 1 week between two CNO injections or two $T_a$/novel cage conditions in the same mice. All recordings were performed in an environmental chamber (Powers scientific, USA) where $T_a$ was controlled within 0.5 °C precision.

**Histology.** Upon completion of data collection, mice in the optogenetic stimulation experiments underwent a 1 Hz stimulation protocol for 2 h and were then deeply anesthetized (chloral hydrate; 700 mg/kg body weight) and transcardially perfused with PBS followed by 10% formalin. Mice in the chemogenetic experiments received CNO (0.3 mg/kg, i.p.) and were perfused after 3 h. Brains were cut into 3 series of 40 μm sections with one series immunolabeled for cFos (as a marker of neuronal activity; 1° antibody—rabbit (Rb) anti-cFos; Oncogene Sciences; cat. no: 4188; 1:30000 dilution) and DsRed (to label hM3Dq-mCherry or ChR2-mCherry-expressing neurons; 1° antibody-Rb-Anti-DsRed; Clontech, USA; cat.no: 632496; 1:10,000 dilution)[30,35]. Similarly, the brain sections from mice in the optogenetic inhibition experiments were labeled for cFos and GFP (to label the ArchT-expressing neurons; 1° antibody-Rb-Anti-GFP; Thermo Fisher Scientific cat. no: A11122; 1:10,000 dilution). cFos immunoreactivity was visualized with 0.06% DAB solution with 0.01% hydrogen peroxide, 0.01% nickel ammonium sulphate and 0.005% cobalt chloride, resulting in black nuclear staining. mCherry/GFP immunoreactivity was then added in a second DAB staining step without nickel and cobalt, resulting in brown labeling of mCherry/GFP-expressing neurons. No mCherry/GFP labeling was observed in areas of the brain that did not receive AAV injections indicating the specificity of the DsRed and GFP antibodies.

**In situ hybridization using RNAScope.** We processed one series of sections for labeling with *GAL* mRNA by in situ hybridization as well as for mCherry/GFP by immunochemistry. We mounted our brain sections on Superfrost Plus slides in RNAs-free conditions and dried them in −20 °C overnight. After warming, we further dried the slides in an oven for 30 min at 40 °C and then performed the RNAscope hybridization using a RNAScope Multiplex Flourescent Reagent Kit V2 (Catalog #323100, Advanced Cell Diagnostics, Hayward, CA). We followed the provided instructions and pretreated the sections with hydrogen peroxide for 20 min at room temperature and then performed an antigen retrieval procedure by placing the slides in a steamer (at >99 °C) for 5 min. We then dehydrated the sections in 100% alcohol and air-dried them for 5 min. Next, we treated the sections with protease reagent (Protease III, RNAscope) for 30 min at 40 °C. After rinsing in sterile water, we incubated the sections in the RNAscope probe for Galanin-C1 (RNAscope® Probe- Mm-Gal;Cat No. 400961 Advanced Cell Diagnostics) for 2 h at 40 °C for the hybridization step. Following the hybridization procedure, we performed three amplification steps at 40 °C (AMP1-FL and AMP2-FL: 30 min each; AMP3-FL: 15 min) and subsequently incubated the sections in HRP blocker for 15 min. We then further incubated the sections in TSA plus Fluorescein/Cy5 fluorophore (Catalog # NEL741001, Perkin Elmer) for 30 min to visualize the *GAL* mRNA (Channel 1 at 488 nm). Since the mCherry or GFP fluorescence from the viral transfection was quenched by these procedures, we performed an additional immunolabeling step to re-establish the mCherry/GFP signals. We incubated the sections in Rabbit anti-DsRed/anti-GFP antibody (1:7500) at 4 °C overnight, then washed them in PBS (2 × 2 min) and incubated them in a secondary antibody (Alexa fluor 555/488 Donkey anti-Rabbit, Thermo Fisher Scientific, Cat. no: A-31572 and A-21206) for 2 hours at room temperature. Finally, we washed the slides one more time before drying and then cover-slipped them with Vectashield mounting medium (Vector Laboratories, Catalog # H-1200).

**Data analysis.** EEG/EMG recordings were scored manually as 10 s epochs (optogenetic experiments) or 12 s epochs (chemogenetic experiments) into wake, NREM or REM sleep using SleepSign software (Kessei Comtec, Japan)[12,13,35]. Percentages of wake, NREM and REM sleep, bout numbers and average bout durations of individual sleep-wake states were calculated for 2 h before, during and after the photostimulations, or every 4 h or 2 h for 24-h after saline or CNO injections depending upon the experiment. EEG power spectra, specifically delta and theta power during individual sleep-wake states in the corresponding periods were also calculated. Mean $T_b$ was calculated every 5 min in all experiments except in Fig. 6 where 2 h means are presented. NREM and REM sleep latencies were calculated as the time taken to the first NREM and REM sleep episode, respectively, from the time of injection. Experimenters scoring sleep-wake data were blinded to the conditions.

**In vitro optogenetic activation and inhibition of VLPO^GAL neurons.** Three to eight weeks after AAV-ChR2-mCherry/AAV-ArchT-GFP virus injections into the VLPO, GAL-cre mice (n = 3 for ChR2; n = 3 for ArchT) were anesthetized with isoflurane (>4%) and brain slices containing the VLPO (250 μm thick) were prepared for in vitro electrophysiological recordings[69]. Whole-cell current clamp

recordings[69] were performed from VLPO^GAL neurons expressing ChR2 or ArchT (identified by mCherry or GFP fluorescence, respectively) using a Multiclamp 700B amplifier (Molecular Devices, Foster City, CA, USA), a Digidata 1322 A interface, and Clampex 9.0 software (Molecular Devices). After achieving stable recordings for 15 min from ChR2-expressing neurons, we photostimulated them using full-field 10 ms flashes of light (~10 mW/mm$^2$, 1 mm beam width; at 0.5, 1, 2, 4, 8, and 16 Hz frequencies) from a 5 W LUXEON blue light-emitting diode (470 nm wavelength; #M470L2-C4; Thorlabs, Newton, NJ, USA) coupled to the epi-fluorescence pathway of the microscope. For ArchT-expressing neurons, we used full-field 5 min light flashes (2.4 mW/mm$^2$) from a 880 mW LUXEON yellow light-emitting diode (565 nm wavelength; #M565L3; Thorlabs, Newton, NJ, USA). We recorded in current clamp mode using K-gluconate-based pipette solution containing (in mM): 120 K-gluconate, 10 KCl, 3 MgCl$_2$, 10 HEPES, 2.5 K-ATP, 0.5 Na-GTP (pH 7.2 adjusted with KOH; 280 mOsm). Data were analyzed using Clampfit 10 (Molecular Devices) and IGOR Pro 6 (WaveMetrics, Lake Oswego, OR, USA) and presented as Mean ± SEM[69].

**In vitro chemogenetic activation of VLPO^GAL neurons.** GAL-Cre mice (n = 5) were injected with AAV-hM3Dq into the VLPO and 6 weeks after the injections, VLPO slices were collected and whole-cell current clamp recordings were performed from a total of five neurons[27,35]. After 15 min of stable recordings, artificial cerebrospinal fluid (aCSF) containing CNO (1 μM solution) was perfused through the chamber and recordings continued for another 5 min before washed out by aCSF[27,35].

**Statistical analysis.** All statistical analyses were performed using GraphPad Prism (version 7.03; GraphPad Software, USA). For optogenetic experiments, we compared sleep-wake and $T_b$ data from 2 h during and 2 h immediately after laser stimulations at various frequencies in VLPO^GAL-ChR2 mice and VLPO^GAL-mCherry mice using two-way repeated measures (RM) ANOVAs followed by Sidak's post hoc comparisons; whereas, the EEG Power spectra were analyzed using Mann–Whitney tests. In vitro data were compared using a one-way ANOVA. For chemogenetic experiments, we compared sleep-wake and $T_b$ data after saline and CNO treatment from VLPO^GAL-hM3dq mice and VLPO^GAL-mCherry mice using two-way RM ANOVAs followed by Sidak's post hoc comparisons. Sleep latencies were analyzed using a Wilcoxon matched-pairs signed-rank test. Sample sizes for optogenetic and chemogenetic mouse cohorts were determined after pilot experiments with 2–3 mice revealed the effect size and variance between mice.

## Data availability
The data that supported the findings of this study are available from the corresponding author upon reasonable request.

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

## Acknowledgements

We thank Quan Ha, Minh Ha, Sam Bragg, Sofia Iqbal, and Bushra Anis for excellent technical assistance. This work was supported by the National Institutes of Health Grants

R21-NS074205 and R01-NS088482 (to R.V.), R01-NS091126 (to E.A.), and P01-AG09975, P01-HL095491, and R01-NS085477 (to C.B.S.).

## Author contributions

Conceived and designed the experiments: D.K., C.B.S., and R.V. Performed physiology and behavioral experiments: D.K. and R.V. Performed In vitro electrophysiology experiments: J.C.M., L.L.F., and E.A. Analyzed data: D.K., G.A., C.G., S.S.B., J.C.M., L.L. F., E.A., C.B.S., and R.V. Contributed transgenic mouse: H.M. Wrote the paper: D.K., C. B.S., and R.V. with input from all the authors

## Additional information

**Competing interests:** The authors declare no competing interests.

