## [Peer Review File · Nature Communications]

Reviewers' comments:

Reviewer #2 (Remarks to the Author):

The study by Kroeger et al demonstrates that GAL neurons in the VLPO facilitate NREM sleep and regulate body temperature, supporting the hypothesis that cells in this brain region are sleep promoting and involved in thermoregulation. Overall, the experiments are well done, well analyzed and well described, and they advance our current understanding of sleep-wake and thermoregulatory mechanisms.

Although the presented data represent a thoughtful series of experiments, the manuscript would be significantly strengthened by data showing how loss of VLPO activity affects sleep. The authors need to include data showing how Arch or hM4Di receptor manipulations influence NREM, REM sleep and waking.

Although expression of c-Fos is increased in rats after periods of increased sleep this same experiment appears to be lacking in mice. It would have been very useful and of immediate relevance to the current study to confirm that GAL cells show increased c-Fos after increased sleep. These data would fit nicely with those presented in FigS1, which shows that lay out of GAL cells by showing which of the GAL cell network are active in NREM sleep. The authors have expertise in this area.

How specific are ChR2 and hM3Dq for GAL cells, ie., how much ectopic expression was observed?

The authors report how different stimulation frequencies (i.e., 0.5-16Hz; Fig 2) of GAL cells influence NREM sleep, but only report how 1Hz stimulation (TableS1) influences REM sleep. Please provide data showing how 0.5-16Hz stimulation impacts REM sleep.

In the Discussion it is indicated that: "mice reduced locomotor activity in preparation for sleep, assumed a sleep posture within their nest and transitioned into sleep" after GAL cell stimulation. This information should also be included in the Results section.

The Discussion needs to be reduced in length and more focused. At present, it is rather lengthy and somewhat unfocused.

Reviewer #3 (Remarks to the Author):

In their article, Kroeger et al. study the effect on sleep of activating and inactivating the galanin neurons of the VLPO.

They showed that the optogenetic activation of these neurons at low frequency between 21 and 23h induces sleep. They further showed that their chemogenetic activation at light or

dark onset strongly increases NREM but decreases REM sleep and also induces a strong hypothermia. They also determine the effect obtained at three room temperature, 22, 29 and 36°C. Finally, they did look at the effect on the effect obtained of moving the animal to a new cage.

Overall, the results obtained are very interesting and some of them are unexpected.

Interestingly, they found that optogenetic activation were effective to induce NREM at frequencies of 0.5 to 4Hz but not at 8 or 16Hz frequencies.

They showed "in vitro" that the stimulation was not effective to drive action potentials when using frequencies of 8 and 16Hz in contrast to lower frequencies.

Fitting with these results, they observed an increase in NREM sleep only at lower frequencies and no clear effect at higher frequencies indicating that such frequencies were not effective to activate the cells.

They acknowledge that their results are apparently not fitting with those obtained recently by Chung et al. (Nature, 2017). Indeed, these authors stimulated with optogenetic at 10Hz the galanin neurons of the VLPO and reported a very strong induction of waking (Suppl Fig. 10). In contrast, they obtained an increase in NREM sleep when stimulating the VLPO neurons projecting to the TMN. This is quite puzzling since 85% of the VLPO neurons projecting to the TMN express galanin. It seems unlikely that the effect obtained by Chung is due to the 15% remaining neurons. It might be that the differences between the two studies are due to the fact that the mice and the AAVs used are not the same.

Taking into account these points, what are the new results brought by the present study?

The present study confirms that stimulation by opto and chemogenetic of VLPO galanin neurons induces an increase in NREM sleep.

In addition, it surprisingly shows that chemogenetic activation of these neurons induces a decrease in REM sleep quantities as well as a large decrease in body temperature.

It also shows an interesting effect of the ambient temperature on the effects obtained.

Finally, it shows that the sleep inducing effect overrides the stress induced by putting the animals in a new cage.

These results are quite new and interesting.

Nevertheless, to my point of view, the optogenetic and chemogenetic inhibition of the VLPO galanin neurons is missing in the report. This indeed would tell us whether these neurons are crucial for inducing NREM sleep. Further, Chung et al. did not report the effect of such inhibition and therefore this would be a great addition to show such experiments.

I'm in fact afraid that the authors did the experiments and did not obtain a convincing effect and therefore did not add them. If this is the case, they should be encouraged to present such negative results. Indeed, this would certainly help us to better understand the role of the VLPO galanin neurons. In contrast, if such experiments are never published, we will continue to ask ourselves whether there will be an effect or not and people will continue to try to do such experiments without publishing them.

Minor points

The first part of the results is presenting the projections of the Galanin neurons. This part is

mainly descriptive without quantification and illustrations. I therefore wonder whether it should be removed or extended. It could be interesting to extend it if it brings something new on the projections of these neurons. Is this the case? This is not clearly stated and discussed.

In the "in vitro" experiments, the authors used the term depolarization block for the inactivation observed. Is it really corresponding to a depolarization block classically observed when putting glutamate agonists on a neuron?

Concerning the optogenetic stimulation, it would be nice if the authors had look at the effect of stimulating specifically during waking and NREM sleep. It would indeed be nice to confirm by this mean that the neurons induce NREM and do not prolong it.

The results obtained with 8 and 16Hz are not very well described. It is written that 16Hz photostimulation increased wakefulness in some mice? This is not reliable therefore?

I also wonder whether there is a positive correlation with the drop of temperature and REM sleep decrease after CNO?

In the discussion, it is written that photostimulation of the neurons did not alter REM sleep in contrast to chemoactivation. The problem here is that the optogenetic stimulation was performed during the night, a period with only a small number of REM sleep episodes.

Optogenetic stimulation during the day might give rise to REM inhibition?

Activation of galanin neurons in the ventrolateral preoptic area promotes sleep and heat loss in mice

Daniel Kroeger¹, Gianna Absi¹, Celia Gagliardi¹, Sathyajit Bandaru¹, Joseph Madara², Loris Ferrari¹, Elda Arrigoni¹, Heike Münzberg³, Thomas E Scammell¹, Clifford B Saper^{1*} and Ramalingam Vetrivelan^{1*}

We thank the reviewers for the valuable comments. Those comments helped us to modify the manuscript which we believe is now greatly improved. We have addressed all comments made by both reviewers in this revised manuscript. We have also included the additional experiments (inhibition of galanin neurons) requested by the reviewers. A point-wise response to reviewers' comments is given below. The revised text in the manuscript is in blue font.

Reviewer #2 (Remarks to the Author):

Comment

The study by Kroeger et al demonstrates that GAL neurons in the VLPO facilitate NREM sleep and regulate body temperature, supporting the hypothesis that cells in this brain region are sleep promoting and involved in thermoregulation. Overall, the experiments are well done, well analyzed and well described, and they advance our current understanding of sleep-wake and thermoregulatory mechanisms.

Although the presented data represent a thoughtful series of experiments, the manuscript would be significantly strengthened by data showing how loss of VLPO activity affects sleep. The authors need to include data showing how Arch or hM4Di receptor manipulations influence NREM, REM sleep and waking.

Response

Thank you. We have now included data describing sleep-wake changes with optogenetic inhibition (using ArchT) of VLPO GAL neurons (Page 9; Manuscript Line 211-242; Fig 3 and Supp fig 4). Data from these experiments indicate that photoinhibition of VLPO GAL neurons increases wake and reduces NREM sleep, but does not alter REM sleep. These data are in line with the photoactivation data and further confirm the importance of VLPO GAL neurons in NREM sleep regulation.

Comment

Although expression of c-Fos is increased in rats after periods of increased sleep this same experiment appears to be lacking in mice. It would have been very useful and of immediate relevance to the current study to confirm that GAL cells show increased c-Fos after increased sleep. These data would fit nicely with those presented in FigS1, which shows that lay out of GAL cells by showing which of the GAL cell network are active in NREM sleep. The authors have expertise in this area.

Response

Our previous study (Gaus et al., 2002) has shown that a high percentage of the sleep-active (Fos-expressing) VLPO neurons express mRNA for GAL not only in rats, but also in mice and cats. This information has been included in the Introduction (Page 3; Manuscript Line 54-56). In addition, this study also analyzed sleep-active VLPO neurons in a diurnal species degus and came to similar conclusions. Finally, a homologous galanin mRNA-containing cell group was found in the ventrolateral part of the preoptic area in diurnal and nocturnal monkeys, as well as in humans (Gaus et al., 2002). As our previous work has established the sleep-active nature of GAL neurons in mice and other species, we did not repeat the counting of cFos active cells, but went ahead and performed stimulation and inhibition experiments as reported in the manuscript.

Comment

How specific are ChR2 and hM3Dq for GAL cells, ie., how much ectopic expression was observed?

Response

By double-labeling sections for mCherry (marking hM3Dq/ChR2 expression) or GFP (marking ArchT expression) and Galanin mRNA, we found that about 88-90% of ChR2/ArchT/hM3Dq-expressing neurons were galaninergic, indicating high-specificity of our AAVs. These results are now included in the revised manuscript (Page 22; Manuscript Line 512-514).

Comment

The authors report how different stimulation frequencies (i.e., 0.5-16Hz; Fig 2) of GAL cells influence NREM sleep, but only report how 1Hz stimulation (TableS1) influences REM sleep. Please provide data showing how 0.5-16Hz stimulation impacts REM sleep.

Response

0.5-16 Hz stimulation of VLPO GAL neurons did not produce significant changes in REM sleep. We have now included this data in the manuscript (Page 7; Manuscript line 153-154 and Suppl. Figure 2k).

Comment

In the Discussion it is indicated that: "mice reduced locomotor activity in preparation for sleep, assumed a sleep posture within their nest and transitioned into sleep" after GAL cell stimulation. This information should also be included in the Results section.

Response

We have included this information in the results section of the text (Page 7; Manuscript line 154-155).

Comment

The Discussion needs to be reduced in length and more focused. At present, it is rather lengthy and somewhat unfocused.

Response

We reduced the length of the discussion and made changes (such as restructuring) to improve the clarity and focus.

Reviewer #3 (Remarks to the Author):

Comment

In their article, Kroeger et al. study the effect on sleep of activating and inactivating the galanin neurons of the VLPO.

They showed that the optogenetic activation of these neurons at low frequency between 21 and 23h induces sleep. They further showed that their chemogenetic activation at light or dark onset strongly increases NREM but decreases REM sleep and also induces a strong hypothermia.

They also determine the effect obtained at three room temperature, 22, 29 and 36°C. Finally, they did look at the effect on the effect obtained of moving the animal to a new cage.

Overall, the results obtained are very interesting and some of them are unexpected.

Interestingly, they found that optogenetic activation were effective to induce NREM at frequencies of 0.5 to 4Hz but not at 8 or 16Hz frequencies.

They showed "in vitro" that the stimulation was not effective to drive action potentials when using frequencies of 8 and 16Hz in contrast to lower frequencies.

Fitting with these results, they observed an increase in NREM sleep only at lower frequencies and no clear effect at higher frequencies indicating that such frequencies were not effective to activate the cells.

They acknowledge that their results are apparently not fitting with those obtained recently by Chung et al. (Nature, 2017). Indeed, these authors stimulated with optogenetic at 10Hz the galanin neurons of the VLPO and reported a very strong induction of waking (Suppl Fig. 10). In contrast, they obtained an increase in NREM sleep when stimulating the VLPO neurons projecting to the TMN. This is quite puzzling since 85% of the VLPO neurons projecting to the TMN express galanin. It seems unlikely that the effect obtained by Chung is due to the 15% remaining neurons. It might be that the differences between the two studies are due to the fact that the mice and the AAVs used are not the same.

Taking into account these points, what are the new results brought by the present study?

The present study confirms that stimulation by opto and chemogenetic of VLPO galanin neurons induces an increase in NREM sleep.

In addition, it surprisingly shows that chemogenetic activation of these neurons induces a decrease in REM sleep quantities as well as a large decrease in body temperature.

It also shows an interesting effect of the ambient temperature on the effects obtained. Finally, it shows that the sleep inducing effect overrides the stress induced by putting the animals in a

new cage.

These results are quite new and interesting.

Nevertheless, to my point of view, the optogenetic and chemogenetic inhibition of the VLPO galanin neurons is missing in the report. This indeed would tell us whether these neurons are crucial for inducing NREM sleep. Further, Chung et al. did not report the effect of such inhibition and therefore this would be a great addition to show such experiments.

I'm in fact afraid that the authors did the experiments and did not obtain a convincing effect and therefore did not add them. If this is the case, they should be encouraged to present such negative results. Indeed, this would certainly help us to better understand the role of the VLPO galanin neurons. In contrast, if such experiments are never published, we will continue to ask ourselves whether there will be an effect or not and people will continue to try to do such experiments without publishing them.

Response

We thank the reviewer for positive comments and highlighting the crucial points of the manuscript. We have now included data describing sleep-wake changes with optogenetic inhibition (using ArchT) of VLPO GAL neurons (Page 9; Manuscript Line 211-242; Fig 3 and Supp fig 4). Data from these experiments indicate that photoinhibition of VLPO GAL neurons increases wake and reduces NREM sleep, but does not alter REM sleep. These data are in line with the photoactivation data and further confirm the importance of VLPO GAL neurons in NREM sleep regulation.

Minor points

Comment

The first part of the results is presenting the projections of the Galanin neurons. This part is mainly descriptive without quantification and illustrations. I therefore wonder whether it should be removed or extended. It could be interesting to extend it if it brings something new on the projections of these neurons. Is this the case? This is not clearly stated and discussed.

Response

Specific projections of VLPO GAL neurons were mostly similar to those of earlier experiments using non-specific tracing of VLPO neurons. Hence, we have reduced the length of this section. We thank you for your suggestion.

Comment

In the "in vitro" experiments, the authors used the term depolarization block for the inactivation observed. Is it really corresponding to a depolarization block classically observed when putting glutamate agonists on a neuron?

Response

Yes. The depolarization block after high frequency optogenetic stimulation of VLPO GAL neurons was similar to that generally observed after application of glutamate agonists onto

neurons in vitro.

Comment

Concerning the optogenetic stimulation, it would be nice if the authors had look at the effect of stimulating specifically during waking and NREM sleep. It would indeed be nice to confirm by this mean that the neurons induce NREM and do not prolong it.

Response

Although we did not specifically stimulate VLPO^{GAL} neurons during wake or NREM sleep, we analyzed the data from our 2 h recordings with continuous stimulation, which indicates that activating VLPO^{GAL} neurons with 1 Hz opto-stimulation significantly increased the number of NREM sleep bouts, but did not increase the average duration of NREM bouts. These findings support the idea that stimulation of VLPO^{GAL} neurons may facilitate wake-to-NREM transitions (induce NREM sleep), but may not prolong NREM sleep bouts. This data has been described in the results as well as in the discussion (Page 8; Manuscript Line 176-179, Page 18 Manuscript Line 411- 422 and Supp Table 1).

Comment

The results obtained with 8 and 16Hz are not very well described. It is written that 16Hz photostimulation increased wakefulness in some mice? This is not reliable therefore?

Response

We observed that 8 and 16 Hz stimulations increased wakefulness in 7 out of 10 mice. The difference (between the 7 vs the other 3 mice) can be explained in terms of the spread and extent of the AAV-ChR2 injections (Page 9; Manuscript line 201-204). We found that in mice with wake-promotion after high frequency stimulation the AAV injections included a larger area in the VLPO and the surrounding preoptic structures. It is likely that inhibition (due to depolarization block) of larger proportion of VLPO GAL neurons may be required to evoke wake effects while the inhibition of a partial set of neurons may not produce wake. This is consistent with our previous results from VLPO lesion studies where we showed that bilateral loss of >70% of VLPO neurons was required to produce *major* sleep loss (Lu et al., 2000; Vetrivelan et al., 2012).

Comment

I also wonder whether there is a positive correlation with the drop of temperature and REM sleep decrease after CNO?

Response

We now include data showing a correlation between hypothermia and REM sleep after chemoactivation of VLPO GAL neurons during the light and dark periods (Page 13; Manuscript Line 298-299). We found that hypothermia and REM sleep were negatively correlated during the light phase ($r=0.57$; $P=0.05$; Fig 5k). In other words (reviewer's) words, there is a positive

correlation between drop in body temperature and REM sleep decrease after CNO during light period. Conversely, during the dark period, many animals had very little REM sleep even after saline and hence the correlation between REM sleep decrease and hypothermia after CNO was not *statistically* significant (Fig. 5h).

Comment

In the discussion, it is written that photostimulation of the neurons did not alter REM sleep in contrast to chemoactivation. The problem here is that the optogenetic stimulation was performed during the night, a period with only a small number of REM sleep episodes. Optogenetic stimulation during the day might give rise to REM inhibition?

Response

We did not find any major changes in REM sleep even when VLPO GAL neurons were stimulated during the daytime (photostimulation at 1 Hz for 2 h starting at 9:00; Figure below). As this finding is not different from REM sleep changes after photoactivation during dark period (presented in the manuscript), we have not included this data/figure in the manuscript.

REVIEWERS' COMMENTS:

Reviewer #2 (Remarks to the Author):

Thank you for addressing my comments. You've done an excellent job doing this. Your manuscript is much improved! A very interesting and important contribution to the field of neuroscience and to sleep biology. I have no more comments.

Reviewer #3 (Remarks to the Author):

The authors well answered to my requests and the addition of the inhibitory experiment is convincing.